# Condition Errors Refinement in Autoregressive Image Generation with Diffusion Loss

**Yucheng Zhou**[*], **Hao Li**[*], **Jianbing Shen**[✉]
SKL-IOTSC, CIS, University of Macau
`yucheng.zhou@connect.um.edu.mo, jianbingshen@um.edu.mo`

## Abstract

Recent studies have explored autoregressive models for image generation, with promising results, and have combined diffusion models with autoregressive frameworks to optimize image generation via diffusion losses. In this study, we present a theoretical analysis of diffusion and autoregressive models with diffusion loss, highlighting the latter's advantages. We present a theoretical comparison of conditional diffusion and autoregressive diffusion with diffusion loss, demonstrating that patch denoising optimization in autoregressive models effectively mitigates condition errors and leads to a stable condition distribution. Our analysis also reveals that autoregressive condition generation refines the condition, causing the condition error influence to decay exponentially. In addition, we introduce a novel condition refinement approach based on Optimal Transport (OT) theory to address "condition inconsistency". We theoretically demonstrate that formulating condition refinement as a Wasserstein Gradient Flow ensures convergence toward the ideal condition distribution, effectively mitigating condition inconsistency. Experiments demonstrate the superiority of our method over diffusion and autoregressive models with diffusion loss methods.

## 1 Introduction

Diffusion models have demonstrated remarkable performance in image generation and have been widely adopted across various visual generative tasks (Wei et al., 2024; Rombach et al., 2022; Saharia et al., 2022). Recently, due to the impressive reasoning capabilities exhibited by large language models (LLMs), autoregressive modeling has garnered significant attention. Consequently, some studies are exploring autoregressive frameworks for image and video generation, aiming to integrate them with LLMs to build more powerful multimodal models (Sun et al., 2024b).

Recent advancements in autoregressive image generation have shown performance comparable to diffusion models (Sun et al., 2024a; Tian et al., 2024; Zhou et al., 2025a). However, most autoregressive image generation methods rely on Vector Quantized Variational Autoencoders (VQ-VAEs (Rombach et al., 2022)) to encode visual content into discrete tokens for next-token prediction modeling. (Li et al., 2024a) indicate that VQ-based image generation is sensitive to gradient approximation strategies and suffers from quantization errors, and propose diffusion loss for autoregressive image generation, effectively pursuing autoregressive image generation without VQ. Nevertheless, a comparative analysis between Conditional diffusion modeling and autoregressive modeling with diffusion loss remains underexplored.

In this study, we investigate the differences between autoregressive modeling with diffusion loss and conditional diffusion modeling. Firstly, we delve into the theoretical underpinnings of patch denoising optimization in autoregressive models for condition error correction. We theoretically prove that, under standard assumptions of Markov property and Gaussian noise in diffusion modeling, the iterative patch denoising approach leads to a stable condition distribution. Furthermore, our analysis

---

[*]Equal Contribution.

[✉]Corresponding Author. This work was supported by the National Natural Science Foundation of China (No. 624B2002), the Science and Technology Development Fund of Macau SAR (FDCT) under grants 0134/2025/RIA2 and 0102/2023/RIA2, and the Jiangyin Hi-tech Industrial Development Zone under the Taihu Innovation Scheme (EF2025-00003-SKL-IOTSC).

reveals the crucial behavior of the conditional probability gradient, showing its attenuation as the condition stabilizes. Our theoretical exploration demonstrates that patch denoising in autoregressive modeling effectively mitigates condition errors and consequently contributes to improved conditional generation quality in diffusion modeling. In addition, we theoretically demonstrate that the sequence of condition variables generated by an autoregressive process effectively refines the condition, leading to a reduction in the gradient norm of the conditional probability distribution. Specifically, we demonstrate that the influence of the condition on the outcome, quantified by the gradient norm, decays exponentially towards a stationary value as the autoregressive iteration progresses.

Building upon these theoretical insights, we further analyze the issue of "condition inconsistency" in autoregressive condition generation, demonstrating how extraneous information accumulates and hinders optimal patch generation. To address this, we introduce a novel condition refinement approach grounded in Optimal Transport (OT) theory. We theoretically prove that formulating condition refinement as a Wasserstein Gradient Flow leads to convergence towards the ideal condition distribution, effectively mitigating condition inconsistency and ultimately enhancing the quality of patch generation within diffusion models.

In the experiments, we compare our method against other diffusion and autoregressive models with diffusion loss on ImageNet (Li et al., 2024b). Results show the superiority of our method over these competitors. We also analyze the denoising process to demonstrate the effectiveness of our method in condition refinement. Our main contributions and findings are as follows:

- We theoretically prove that patch denoising optimization in autoregressive models mitigates condition errors and elucidates the attenuation behavior of the conditional probability gradient as the condition stabilizes.

- We theoretically establish the efficacy of autoregressive condition refinement, quantifying the exponential decay of the condition's influence on the outcome as autoregressive iteration progresses to a stationary value.

- We propose a condition refinement method based on Optimal Transport theory, and theoretically prove that formulating it as a Wasserstein Gradient Flow ensures convergence towards the ideal condition distribution.

- Experiments demonstrate our method's superiority over other competitors. Extensive analysis shows the effectiveness of our method in condition refinement.

## 2 PRELIMINARIES

**Diffusion Modeling.** Diffusion models are generative frameworks that consist of a forward process. The forward (diffusion) process is a Markov chain that transforms data $x_0$ into Gaussian noise $x_T$ through a sequence of Gaussian transitions:

$$q(x_{1:T}|x_0) = \prod_{t=1}^{T} q(x_t|x_{t-1}), \quad q(x_t|x_{t-1}) = \mathcal{N}(x_t; \sqrt{1-\beta_t}x_{t-1}, \beta_t\mathbf{I}), \tag{1}$$

where $\beta_{t_{t=1}}^T$ is a predefined variance schedule with $0 < \beta_1 < \cdots < \beta_T < 1$. The reverse (denoising) process reconstructs $x_0$ from $x_T$ via:

$$p_\theta(x_{0:T}) = p(x_T)\prod_{t=1}^{T} p_\theta(x_{t-1}|x_t), \quad p_\theta(x_{t-1}|x_t) = \mathcal{N}(x_{t-1}; \mu_\theta(x_t,t), \Sigma_\theta(x_t,t)), \tag{2}$$

where $\mu_\theta$ and $\Sigma_\theta$ are predicted by a neural network. Since the true posterior $q(x_{t-1}|x_t)$ is intractable, it is approximated using $q(x_{t-1}|x_t, x_0)$ during training.

The model is trained to maximize data likelihood, which is approximated via a variational lower bound. This can be reformulated as a score-matching problem, e.g.,

$$\mathbb{E}x_t \sim p_t(x_t) \left[ |\nabla x_t \log p_t(x_t) - s_\theta(x_t,t)|^2 \right]. \tag{3}$$

**Autoregressive Modeling.** Autoregressive (AR) models are generative frameworks that sequentially predict each element in a data sequence by conditioning on all preceding elements. These models assume that each data point $x_i$ depends only on the prior points $x_{<i} = \{x_1, x_2, \ldots, x_{i-1}\}$. The conditional and joint probabilities can be expressed as:

$$p(x) = p(x_1, x_2, \ldots, x_n) = \prod_{i=1}^{n} p(x_i | x_{<i}) \tag{4}$$

The generation process starts from $x_1$ and proceeds sequentially to $x_n$, with each step conditioned on all previously generated elements. Related work is in Appendix A.

## 3 THEORETICAL ANALYSIS ON AUTOREGRESSIVE IMAGE MODELING WITH DIFFUSION LOSS

### 3.1 DIFFERENCE OF DIFFUSION MODELS

Diffusion models demonstrate exceptional capabilities in generating high-quality visual content. Recently, autoregressive modeling integrated with diffusion loss has shown significant potential in image generation. We will elucidate the differences between standard conditional diffusion modeling and autoregressive modeling with diffusion loss.

**Conditional Diffusion Modeling.** In traditional conditional diffusion models, the reverse process is conditioned on a single, static condition $c$. This can be formally expressed as:

$$x_{t-1} \sim p(x_{t-1} | x_t, c), \tag{5}$$

where $c$ represents a global condition that influences every step of the denoising trajectory. When dealing with images, we can extend this to an individual patch $x_i$:

$$x_{i,t-1} \sim p(x_{i,t-1} | x_{j,t}, c), \quad \forall j \in \{1, \ldots, n\}, \tag{6}$$

where $n$ represents the number of patches. Each patch is denoised based on the same shared condition $c$, irrespective of its position within the image.

**Autoregressive Modeling with Diffusion Loss.** Autoregressive modeling with diffusion loss allows the condition to evolve autoregressively. Instead of a fixed condition $c$, a sequence of conditions $\{c_i\}$ depends on preceding conditions $\{c_{<i}\}$ including the initial condition $c_0$, i.e.,

$$c_t \sim p(c_t | c_{<i}, c_0), \tag{7}$$

where $c_{<i}$ denotes all conditions up to time $i - 1$, i.e., $\{c_0, c_1, \ldots, c_{i-1}\}$. For each patch generation $x_i$, the reverse process is still a denoising process but is guided by the dynamic condition $c_i$, i.e.,

$$x_{i,t-1} \sim p(x_{i,t-1} | x_{i,t}, c_i), \tag{8}$$

where $c_i$ represents the condition for $i$-th patch. After generating $x_i$, it is passed as input to the autoregressive model along with the history of conditions $\{c_{<i+1}\}$, enabling the prediction of the subsequent condition $c_{i+1}$.

### 3.2 CONDITIONAL DENOISING MODEL ERROR DEFINITION

**Conditional Score Matching as an Upper Bound.** Score matching is central to training diffusion models, and its loss is linked to the Wasserstein distance between generated and real data (Kwon et al., 2022). Conditional score matching refines this by incorporating conditioning. Understanding how conditional score matching relates to standard score matching is key to justifying its use. This section establishes that the standard score matching loss is upper-bounded by its conditional counterpart. This result supports the use of conditional score matching, suggesting it might lead to a more controlled training process.

**Theorem 1** (Conditional Score Matching Upper Bound). *The standard score matching loss is upper-bounded by the conditional score matching loss:*

$$\mathbb{E}_{\mathbf{x}_t \sim p_t(\mathbf{x}_t)} \left[ \| \nabla_{\mathbf{x}_t} \log p_t(\mathbf{x}_t) - s_\theta(\mathbf{x}_t, t) \|^2 \right] \tag{9}$$

$$\leq \mathbb{E}_{c \sim p_c(c), \mathbf{x}_t \sim p_t(\mathbf{x}_t | c)} \left[ \| \nabla_{\mathbf{x}_t} \log p_t(\mathbf{x}_t | c) - s_\theta(\mathbf{x}_t, t) \|^2 \right]$$

*See Appendix C for the proof, which uses the law of total probability and Jensen's inequality.*

The conditional score matching loss serves as an upper bound for the standard score matching loss. Consequently, minimizing the conditional score matching loss indirectly constrains the standard score matching loss from above.

**Error in Conditional Score Matching.** To analyze the error in conditional score matching, we build upon the score matching error definition from (Li & Yan, 2024):

$$\epsilon_{\text{score}}^2 := \frac{1}{T} \sum_{t=1}^{T} \mathbb{E}_{X \sim q_t} \left[ \|s_t(X) - s_t^*(X)\|_2^2 \right]. \tag{10}$$

To understand how conditioning affects the error structure, we need to decompose the conditional and unconditional score matching losses. Expanding these losses into their component terms allows us to identify and analyze the specific contributions of conditioning to the overall error.

**Lemma 1** (Expansion of Score Matching Loss). *Expanding the square term in the score matching loss, we get:*

$$\mathbb{E}_{x_t \sim p_t(x_t)} \left[ \|\nabla_{x_t} \log p_t(x_t) - s_\theta(x_t, t)\|^2 \right]$$
$$= \mathbb{E}_{x_t \sim p_t(x_t)} \left[ \|\nabla_{x_t} \log p_t(x_t)\|^2 + \|s_\theta(x_t, t)\|^2 - 2 \langle s_\theta(x_t, t), \nabla_{x_t} \log p_t(x_t) \rangle \right]. \tag{11}$$

*Similarly, for conditional score matching:*

$$\mathbb{E}_{c, x_t \sim p_c(c) p_t(x_t|c)} \left[ \|\nabla_{x_t} \log p_t(x_t|c) - s_\theta(x_t, t)\|^2 \right]$$
$$= \mathbb{E}_{c, x_t \sim p_c(c) p_t(x_t|c)} \left[ \|\nabla_{x_t} \log p_t(x_t|c)\|^2 + \|s_\theta(x_t, t)\|^2 - 2 \langle s_\theta(x_t, t), \nabla_{x_t} \log p_t(x_t|c) \rangle \right] \tag{12}$$

*This expansion separates the loss into terms related to the true score, the estimated score, and their interaction, facilitating a more granular error analysis (Detailed Proof in Appendix D.1).*

**Definition 1** (Conditional Error Term $\epsilon_c$). To specifically measure the impact of conditioning on the true score's magnitude, we define the conditional error term $\epsilon_c$ as the change in the expected squared norm of the true score due to conditioning, relative to the unconditional case:

$$\epsilon_c = \frac{1}{T} \sum_{t=1}^{T} \mathbb{E}_{x_t \sim p_t(x_t)} \left[ \mathbb{E}_{c \sim p_t(c|x_t)} \left[ \|\nabla_{x_t} \log p_t(x_t|c)\|^2 \right] - \|\nabla_{x_t} \log p_t(x_t)\|^2 \right] \tag{13}$$

This term quantifies how much the expected squared norm of the true score changes when we move from unconditional to conditional score matching. A positive $\epsilon_c$ would suggest that conditioning increases the magnitude of the true score, potentially indicating a more complex or refined score function (Detailed Proof in Appendix D.2).

**Definition 2** (Simplified Conditional Error Term $\bar{\epsilon}_c$). For a simpler metric focused purely on the magnitude of the conditional true score, we define the simplified conditional error term $\bar{\epsilon}_c$:

$$\bar{\epsilon}_c = \frac{1}{T} \sum_{t=1}^{T} \mathbb{E}_{c \sim p_c(c), \mathbf{x}_t \sim p_t(\mathbf{x}_t|c)} \left[ \|\nabla_{x_t} \log p_t(x_t|c)\|^2 \right] \tag{14}$$

$\bar{\epsilon}_c$ directly measures the expected squared norm of the conditional score. Analyzing $\bar{\epsilon}_c$, along with $\epsilon_c$, will help understand the behavior of the true score in conditional settings (Detailed Proof in Appendix D.3).

## 3.3 CONDITIONAL CONTROL TERM ANALYSIS.

We first investigate the uniqueness of the conditional control term under standard diffusion assumptions and Classifier-Free guidance. As described in Classifier-Free guidance (Ho & Salimans, 2022; Liu et al., 2023), the conditional reverse diffusion process for sampling is given by:

$$p(x_{t-1}|x_t) = \mathcal{N}(x_{t-1}; \mu(x_t) + \sigma_t^2 \nabla_{x_t} \log p(c|x_t), \sigma^2 I)$$
$$x_{t-1} = \mu(x_t) + \sigma_t^2 \nabla_{x_t} \log p(c|x_t) + \sigma_t \epsilon, \epsilon \sim \mathcal{N}(0, 1) \tag{15}$$

This shows that conditional control introduces an additional term $\sigma_t^2 \nabla_{x_t} \log p(c|x_t)$ to the mean of the reverse process, compared to the unconditional diffusion sampling. To understand the impact of this conditional term, we define $f(c_i) = \|\nabla_{x_t} \log p_t(x_t|c_i)\|^2$. We hypothesize that the difference between the expected value of $f(c_i)$ and the expected value of the unconditional score norm isolates the contribution of this conditional control term. This is formalized in the following lemma:

**Lemma 2** (Uniqueness of Conditional Control Term). *Under standard diffusion assumptions and Classifier-Free guidance, the difference between the expected squared norm of the conditional score function and the unconditional score function isolates the contribution of the conditional control term. Let $f(c_i) := \|\nabla_{x_t} \log p_t(x_t|c_i)\|^2$. Then,*

$$\mathbb{E}[f(c_i)] - \mathbb{E}\|\nabla_{x_t} \log p_t(x_t)\|^2 = \mathbb{E}\|\sigma_t^2 \nabla_{x_t} \log p(c|x_t)\|^2 \tag{16}$$

*where the expectation is taken over $x_t \sim p_t(x_t)$ and $c \sim p_c(c)$. This lemma indicates that $f(c_i)$, through its expected difference with the unconditional score norm, precisely captures the impact of the conditional guidance term $\sigma_t^2 \nabla_{x_t} \log p(c|x_t)$ in the diffusion denoising process. (Proof in Appendix E)*

## 3.4 CONDITION REFINEMENT THROUGH PATCH DENOISING.

Building upon the observation that incorporating conditions can amplify errors in diffusion models, we propose an optimization strategy focused on refining the condition using patch-related corrections. Specifically, we introduce a mechanism where information from each newly generated patch is propagated to the condition of the subsequent patch through an iterative update process, $c_{i+1} = \mathcal{T}(c_i)$. This autoregressive approach aims to refine the condition during the denoising process iteratively.

To formalize our approach, we first establish the foundational assumptions under which our model operates.

**Assumption 1** (Markov Property Assumption). The reverse diffusion process adheres to the Markov property, where each state $x_{t-1}$ is conditionally dependent only on the current state $x_t$.

**Assumption 2** (Gaussian Distribution Assumption). The conditional probability distribution $p_t(x_{t-1}|x_t)$ in the reverse diffusion process is assumed to be Gaussian, expressed as: $p_t(x_{t-1}|x_t) = \mathcal{N}(x_{t-1}; \mu(x_t), \sigma_t^2 I)$.

**Assumption 3** (Small Variance Assumption). As the number of time steps $T$ becomes sufficiently large, the variance $\sigma_t^2$ of the conditional distribution $p_t(x_{t-1}|x_t)$ is assumed to be sufficiently small, approaching zero as $T$ increases. Furthermore, for simplicity, we approximate the variance at each step to be equal and denote it as $\sigma^2$.

With the above assumptions, we model the patch refinement process as an iterative update to the condition:

$$c_{i+1} = \mathcal{T}(c_i), \quad i \in \mathbb{N} \tag{17}$$

where $c_{i+1}$ represents the condition at the $(i+1)$-th iteration, corresponding to the refinement based on the $i$-th patch. $\mathcal{T}$ is a diffusion function that governs the transition from the current condition state $c_i$ to the next state $c_{i+1}$, encapsulating the information propagation from the generated patch to the subsequent condition. The index $i \in \mathbb{N}$ denotes the iteration step, analogous to discrete time steps. The condition update process forms a discrete-time Markov chain, as the future state $c_{i+1}$ depends only on the present state $c_i$: $P(c_{i+1}|c_i, c_{i-1}, \ldots, c_0) = P(c_{i+1}|c_i)$.

From the Gaussian expansion norm in Equation equation 15, we observe that the probability distribution of $x_t$ is influenced by $x_t$ itself. Our primary goal is to understand the trajectory of the conditional probability gradient as the condition $c_i$ iteratively refines through the diffusion reverse process. Therefore, we proceed to analyze how the conditional probability gradient evolves with the iterations of $c_i$ within a standard normal conditional distribution setting.

**Proposition 1** (Condition Refinement via Patch Denoising). *In the diffusion denoising process, autoregressively refining the condition through patch-related corrections using the iterative update $c_{i+1} = \mathcal{T}(c_i)$ leads to improved conditional generation quality.(Detailed Proof in Appendix F)*

### 3.5 AUTOREGRESSIVE MODELING CAN REFINE CONDITION

An autoregressive process is defined as follows:

$$c_{i+1} = \sum_{j=0}^{i} a_j c_j + \varepsilon_{i+1}, \quad i \in \mathbb{N} \tag{18}$$

**Assumption 4** (Basic Assumptions for Autoregressive Process). We provide a set of standard assumptions:

1. $\sum_{i=0}^{\infty} |a_i| < \infty$ and $\sup_{i \in \mathbb{N}} |a_i| < 1$, i.e., the sequence $\{a_n\}$ is convergent.

2. $\{\varepsilon_i\}_{i=1}^{\infty}$ are independent and identically distributed, following $\mathcal{N}(0, \sigma^2)$.

3. $p_t(x_t|c_i)$ has continuous second-order derivatives with respect to $x_t$.

4. $\|\nabla_{x_t}^2 p_t(x_t|c_i)\| \leq K$ for some constant $K > 0$ uniformly holds, i.e., is bounded.

5. $(\mathcal{X}, \|\cdot\|)$ is a separable complete metric space.

**Lemma 3** (Markov Property (Meyn & Tweedie, 2012) (Bellet, 2006)). *Under Assumption 4, by defining the state vector $\mathbf{c}_i = (c_i, c_{i-1}, \ldots, c_{i-p+1})^{\top}$, the sequence $\{\mathbf{c}_i\}_{i \in \mathbb{N}}$ forms a strong Markov chain. Specifically:*

1. *The transition probability kernel $P(\mathbf{c}_{i+1} \in \cdot|\mathbf{c}_i)$ on the augmented state space satisfies the Feller property.*

2. *There exists a unique invariant probability measure $\pi \in \mathcal{P}(\mathcal{X}^p)$ such that*

$$\pi P = \pi, \text{ i.e. } \int_{\mathcal{X}^p} P(A|\mathbf{c})\pi(d\mathbf{c}) = \pi(A), \ \forall A \in \mathcal{B}(\mathcal{X}^p) \tag{19}$$

3. *There exist constants $C > 0$ and $\rho \in (0, 1)$ such that for any initial distribution $\mu \in \mathcal{P}(\mathcal{X}^p)$:*

$$\|\mu P^n - \pi\|_{TV} \leq C\rho^n \|\mu - \pi\|_{TV}, \quad \forall n \in \mathbb{N}_0 \tag{20}$$

*where $\|\cdot\|_{TV}$ denotes the total variation norm, and $P^n$ denotes the $n$-step transition probability kernel.*

*In particular, for any $n \in \mathbb{N}_0$, we have:*

$$\|\mathcal{L}(\mathbf{c}_n) - \pi\|_{TV} \leq C\rho^n \tag{21}$$

*where $\mathcal{L}(\mathbf{c}_n)$ denotes the distribution of $\mathbf{c}_n$. Proof can be found in Appendix J.*

**Lemma 4** (Regularity of Conditional Probability (Durrett, 1996)). *Under Assumption 4, there exist constants $\delta, L > 0$ such that:*

1. *$p_t(x_t|c_i) \geq \delta$ for all $(x_t, c_i) \in \mathcal{X} \times \mathcal{X}$.*

2. *$\|\nabla_{x_t} p_t(x_t|c_1) - \nabla_{x_t} p_t(x_t|c_2)\| \leq L\|c_1 - c_2\|$ for all $x_t, c_1, c_2 \in \mathcal{X}$.*

*Proof can be found in Appendix I.*

**Lemma 5** (Bounded Derivative Theorem). *On a fixed bounded closed interval $[a, b]$, if the second derivative is bounded, then there exist constants $M_1, M_2 > 0$ such that:*

1. *$\sup_{x_t} \|\nabla_{x_t} p_t(x_t|c)\| < M_1$, i.e., the first derivative is bounded.*

2. *$\sup_{x_t} |p_t(x_t|c)| < M_2$, i.e., the original function is bounded.*

*Proof can be found in Appendix M.*

**Theorem 2** (Descent of Gradient Norm in Autoregressive Process). *Under Assumptions 4 and Lemmas 3, 4, 5, there exist constants $M > 0$ and $\beta \in (0, 1)$ such that for any $x_t \in \mathcal{X}$ and $i \in \mathbb{N}_0$:*

$$\|\nabla_{x_t} \log p_t(x_t|c_i)\| \leq M\beta^i + m \tag{22}$$

*where $m$ is a constant representing the stationary gradient norm (Proof in Appendix G).*

## 4 AUTOREGRESSIVE CONDITION OPTIMIZATION

Although autoregressive methods provide contextual information, they inevitably accumulate extraneous noise, leading to "condition inconsistency".

**Why Optimal Transport?** We employ Optimal Transport (OT) to rectify this distributional drift for three theoretical reasons:

1. **Geometric Correction:** Unlike overlap-based metrics (e.g., KL divergence), OT quantifies the geometric cost required to transform the noisy generated distribution back to the ideal one.

2. **Least Action Principle:** Formulating the refinement as a Wasserstein Gradient Flow identifies the optimal path to eliminate inconsistency while preserving valid semantic information.

3. **Convergence:** The framework guarantees theoretical convergence to the stationary ideal distribution, effectively acting as a mathematically grounded "denoising" step for the condition.

The full algorithm is provided in Appendix L.

### 4.1 CONDITION INCONSISTENCY IN AUTOREGRESSIVE GENERATION

The autoregressive condition generation process, as defined by Equation equation 7, sequentially constructs conditions, aiming to capture contextual dependencies. However, this sequential nature can lead to conditions that are not only influenced by relevant preceding patches but also by accumulated information that is extraneous to generating the current patch. This phenomenon, which we term "condition inconsistency", arises because the autoregressive process, while capturing dependencies, does not inherently guarantee that each generated condition $c_i$ is optimally focused on information strictly necessary for the corresponding patch $x_i$.

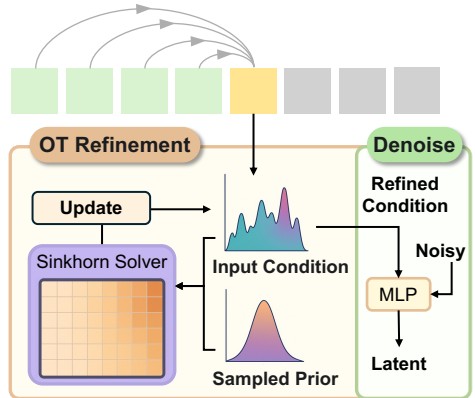

Figure 1: The autoregressive model predicts an initial condition, which is processed by the OT Refinement module using a sampled prior derived from Algorithm 1. The resulting refined condition then guides the Denoise MLP for latent generation.

**Lemma 6** (Condition Information Inconsistency and Extraneous Information Accumulation). *Let $c_i = \Phi_\theta(c_{i-1}) + \Gamma_\theta(\epsilon_i)$ be the autoregressively generated condition for patch $x_i$, and $c_i^* = \pi_{\mathcal{I}_i^*}(c_i)$ be its projection onto the minimal sufficient information subspace $\mathcal{I}_i^*$ (derived from $x_{<i}$). The generated $c_i$ inherently contains an extraneous information component $\eta_i = c_i - c_i^*$. This component $\eta_i$ is generally non-zero (i.e., $\mathbb{E}[\|\eta_i\|_2^2] > 0$), accumulating from $(I - \pi_{\mathcal{I}_i^*})\Phi_\theta(c_{i-1})$ and noise components outside $\mathcal{I}_i^*$. The actual conditional distribution $p(x_i|c_i)$ deviates from the ideal $p(x_i|c_i^*)$, and the conditional score $\nabla_{x_t} \log p(x_t|c_i)$ is perturbed from its optimal form under $c_i^*$.*

*Proof.* Let $\mathcal{I}_i^* \subseteq \mathbb{R}^d$ denote the minimal sufficient information subspace for generating patch $x_i$, and $\pi_{\mathcal{I}_i^*}(\cdot)$ be the orthogonal projection onto this subspace. The ideal condition $c_i^*$ satisfies:

$$c_i^* = \pi_{\mathcal{I}_i^*}(c_i) \quad \text{where} \quad \mathcal{I}_i^* = \text{span}\{f_k(x_{<i})\}_{k=1}^K \tag{23}$$

for some basis functions $\{f_k\}$ encoding relevant dependencies from preceding patches $x_{<i}$. The autoregressive condition generation follows a Markov process:

$$c_i = \Phi_\theta(c_{i-1}) + \Gamma_\theta(\epsilon_i) \tag{24}$$

where $\Phi_\theta : \mathbb{R}^d \to \mathbb{R}^d$ is the learned transition operator and $\Gamma_\theta$ modulates the noise injection. The extraneous information component $\eta_i$ can be quantified through subspace decomposition:

$$\eta_i = (I - \pi_{\mathcal{I}_i^*})c_i = \sum_{k=K+1}^{\infty} \langle c_i, v_k \rangle v_k \tag{25}$$

where $\{v_k\}$ forms an orthonormal basis for $\mathbb{R}^d$ with the first $K$ vectors spanning $\mathcal{I}_i^*$. The $\ell^2$-norm of extraneous information $\|\eta_i\|_2$ satisfies:

$$\mathbb{E}[\|\eta_i\|_2^2] = \mathbb{E}\left[\left\|(I - \pi_{\mathcal{I}_i^*})\Phi_\theta(c_{i-1})\right\|_2^2\right] + \mathrm{tr}(\Gamma_\theta\Gamma_\theta^\top) \tag{26}$$

The first term represents propagated extraneous information from previous conditions, while the second term quantifies newly introduced noise. For the denoising process $\mathcal{D}_t$ at timestep $t$, the conditional score function becomes perturbed:

$$\nabla_{x_t} \log p(x_t|c_i) = \underbrace{\nabla_{x_t} \log p(x_t|c_i^*)}_{\text{Ideal score}} + \underbrace{\mathbf{J}_{\eta_i} \nabla_{c_i} \log p(x_t|c_i)}_{\text{Perturbation term}} \tag{27}$$

where $\mathbf{J}_{\eta_i}$ is the Jacobian of the perturbation. The extraneous information induces an $\mathcal{O}(\|\eta_i\|_2)$ deviation from the ideal denoising trajectory. The accumulated effect over $N$ patches yields total inconsistency:

$$\mathcal{E}_{\text{total}} = \sum_{i=1}^N \mathbb{E}\left[\mathrm{OT}_\lambda(p(x_i|c_i), p(x_i|c_i^*))\right] \tag{28}$$

where $\mathrm{OT}_\lambda$ denotes the Sinkhorn divergence with regularization parameter $\lambda$. This completes the proof of condition information inconsistency. □

## 4.2 Optimal Transport for Condition Refinement via Wasserstein Gradient Flow

Building upon the condition inconsistency analysis in Lemma 6, we present a principled solution through optimal transport theory. Our approach establishes direct connections between the Wasserstein gradient flow framework and condition refinement in autoregressive generation.

**Proposition 2** (Optimal Transport as Wasserstein Gradient Flow). *The condition refinement process can be formulated as a Wasserstein gradient flow that minimizes:*

$$\mathcal{F}(P_c) := W_2^2(P_c, P_{c^*}) + \lambda\mathbb{E}_{c\sim P_c}[\|c - \mathcal{T}^{-1}(x)\|^2] \tag{29}$$

*where $P_{c^*}$ denotes the ideal condition distribution and $\mathcal{T}^{-1}$ represents the inverse process of information accumulation in Equation equation 24. The solution admits an implementable discrete-time scheme through JKO iterations (Jordan et al., 1998):*

$$P_c^{(k+1)} = \arg\min_P W_2^2(P, P_c^{(k)}) + \eta_k\mathcal{F}(P) \tag{30}$$

*Proof.* Let $\mathcal{P}_2(\mathbb{R}^d)$ denote the space of probability measures with finite second moments. We consider the energy functional:

$$\mathcal{F}(P) = \frac{1}{2}W_2^2(P, P_{c^*}) + \lambda\mathbb{E}_{c\sim P}[\phi(c)] \tag{31}$$

where $\phi(c) = \|c - \mathcal{T}^{-1}(x)\|^2$ encodes the inverse process regularization. The Wasserstein gradient flow $\partial_t P_t = -\nabla_{W_2}\mathcal{F}(P_t)$ can be discretized via the Jordan-Kinderlehrer-Otto (JKO) scheme:

$$P^{(k+1)} = \arg\min_P \left\{W_2^2(P, P^{(k)}) + 2\eta_k\mathcal{F}(P)\right\} \tag{32}$$

Substituting our specific energy functional yields the update rule in Proposition 2. The first term maintains proximity to previous iterates while the second term drives the distribution toward both the ideal condition and inverse-process consistency.

The optimal transport plan between $P^{(k)}$ and $P^{(k+1)}$ corresponds to the McCann interpolant:

$$c^{(k+1)} = c^{(k)} - \eta_k\left[\nabla W_2^2(\cdot, P_{c^*})|_{c^{(k)}} + \lambda\nabla\phi(c^{(k)})\right] \tag{33}$$

Implementing this requires solving the regularized OT problem:

$$\inf_{\gamma\in\Gamma(P^{(k)}, P_{c^*})} \mathbb{E}_{(c,c')}[\|c - c'\|^2] + \epsilon\mathrm{KL}(\gamma|\pi) \tag{34}$$

where $\pi$ is the independent coupling and $\epsilon$ controls entropy regularization. This leads to the Sinkhorn algorithm implementation described in Proposition 2. □

**Theorem 3** (Convergence of Wasserstein Gradient Flow). *Under the assumptions of Proposition 2, the Wasserstein gradient flow defined by the energy functional $\mathcal{F}(P)$ converges to the ideal condition distribution $P_{c^*}$. Specifically, for any initial distribution $P_c^{(0)} \in \mathcal{P}_2(\mathbb{R}^d)$, the sequence of distributions $\{P_c^{(k)}\}_{k=1}^{\infty}$ generated by the JKO scheme satisfies:*

$$W_2(P_c^{(k)}, P_{c^*}) \leq \rho^k W_2(P_c^{(0)}, P_{c^*}), \tag{35}$$

*where $\rho < 1$ is the contraction rate determined by the regularization parameter $\lambda$ and the step size $\eta_k$.*

*Proof Sketch.* The proof follows from the contractive properties of the Wasserstein gradient flow for convex energy functionals. By the JKO scheme and the regularization term $\lambda \mathbb{E}_{c \sim P_c}[\|c - \mathcal{T}^{-1}(x)\|^2]$, the sequence $\{P_c^{(k)}\}$ forms a Cauchy sequence in the Wasserstein space $\mathcal{P}_2(\mathbb{R}^d)$. The contraction rate $\rho$ arises from the strong convexity of the energy functional $\mathcal{F}(P)$ and Lipschitz continuity of the gradient flow. □

**Remark 1.** The inverse process regularization $\mathcal{T}^{-1}$ directly counters the extraneous information accumulation characterized in Equation equation 26. By Theorem 3, the refinement ensures monotonic improvement in patch generation quality:

$$\mathbb{E}[\text{OT}_\lambda(p(x_i|c_i^{(k)}), p(x_i|c_i^*))] \leq \rho^k \mathbb{E}[\text{OT}_\lambda(p(x_i|c_i^{(0)}), p(x_i|c_i^*))] \tag{36}$$

where $\rho < 1$ quantifies the contraction rate of our OT-based refinement operator.

This theorem ensures that the proposed Wasserstein gradient flow refinement process monotonically reduces the Wasserstein distance between the autoregressive condition distribution $P_c^{(k)}$ and the ideal condition distribution $P_{c^*}$. The contraction rate $\rho$ quantifies the refinement efficiency, with smaller values of $\rho$ indicating faster convergence.

## 5 EXPERIMENTS

### 5.1 EXPERIMENTAL SETTINGS

Our autoregressive model is directly based on GPT-XL, while the denoising module is implemented using the MAR-based denoising module. For the Variational Autoencoder (VAE) component, we use the KL-16 version of LDM (Rombach et al., 2022). Our experiments are conducted on the ImageNet dataset (Deng et al., 2009), with image resolutions set to $256 \times 256$. For evaluation, we adopt Fréchet Inception Distance (FID) (Heusel et al., 2017), Inception Score (IS) (Salimans et al., 2016), as well as Precision and Recall metrics (Dhariwal & Nichol,

Table 1: Comparison of different methods on various metrics on ImageNet 256×256 conditional generation. Baseline (CDM) denotes a baseline of conditional diffusion modeling.

| Method | FID ↓ | IS ↑ | Pre. ↑ | Rec. ↑ |
|---|---|---|---|---|
| LDM-4 (Rombach et al., 2022) | 3.60 | 247.7 | 0.87 | 0.48 |
| U-ViT-H/2-G (Bao et al., 2022) | 2.29 | 263.9 | 0.81 | 0.62 |
| DiT-XL/2 (Peebles & Xie, 2023) | 2.27 | 278.2 | 0.83 | 0.57 |
| DiffiT (Hatamizadeh et al., 2024) | 1.73 | 276.5 | 0.80 | 0.62 |
| MDTv2-XL/2 (Gao et al., 2023) | 1.58 | 314.7 | 0.79 | 0.65 |
| GIVT (Tschannen et al., 2024) | 3.35 | - | 0.84 | 0.53 |
| MAR (Li et al., 2024a) | 1.55 | 303.7 | 0.81 | 0.62 |
| De-MAR (Yao et al., 2025) | 1.47 | 305.8 | 0.83 | 0.62 |
| RAR (Yu et al., 2025) | 1.50 | 306.9 | 0.80 | 0.62 |
| Baseline (CDM) | 3.26 | 259.6 | 0.81 | 0.58 |
| Baseline (AR) | 2.02 | 282.6 | 0.80 | 0.59 |
| MAR (Li et al., 2024a) | 1.55 | 303.7 | 0.81 | 0.62 |
| Ours (AR) | 1.52 | 317.6 | 0.82 | 0.60 |
| Ours (MAR) | **1.31** | **324.2** | 0.81 | 0.63 |

2021). During training, the noise schedule follows a cosine shape and consists of 1000 steps. The learning rate is set to $1 \times 10^{-5}$, with a total of 400 epochs and a batch size of 2048. The models are trained with a 100-epoch linear learning rate warmup. We use an exponential moving average (EMA) in parameters with a momentum of 0.9999.

### 5.2 PERFOMANCE COMPARISON

Table 1 shows a comparison of our method against state-of-the-art approaches on ImageNet $256 \times 256$ conditional generation. Our method achieves the best FID score of **1.52**, outperforming MAR (Li et al., 2024a) (1.55), MDTv2-XL/2 (Gao et al., 2023) (1.58), and DiffiT (Hatamizadeh et al., 2024) (1.73). Based on MAR, it can further reach 1.31. In IS, our method also achieves the highest score. This demonstrates that our model produces samples with higher

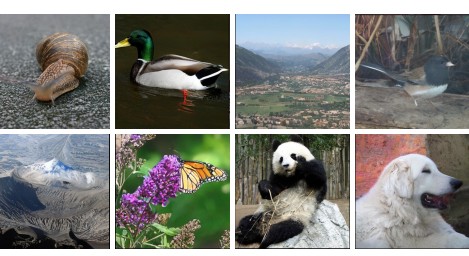

Figure 2: Qualitative results on $256 \times 256$ ImageNet class-conditional generation. These images are generated by Ours.

fidelity and better alignment with the real image distribution. For Precision and Recall, it attains 0.81 and 0.63, respectively, remaining competitive with other methods. Compared to the baseline, our model exhibits significant improvements across all evaluation metrics, highlighting the effectiveness of our approach. Qualitative results are shown in Figure 2.

## 5.3 SCALABILITY ANALYSIS

To investigate the scalability and robustness of our proposed method, we conducted additional evaluations across varying model sizes and higher image resolutions. These experiments aim to verify whether the benefits of our Condition Refinement approach persist as the model capacity increases and the generation task becomes more challenging.

**Scalability across Model Sizes.**   We evaluated our method against the strong baseline MAR (Li et al., 2024a) on ImageNet $256 \times 256$ using three different model scales: 208M, 479M, and 943M parameters. As presented in Table 2, our method consistently outperforms MAR across all model sizes. Notably, the performance gap widens as the model size increases, suggesting that our autoregressive condition optimization effectively leverages larger capacities for superior generation quality.

Table 2: Comparison of scalability across different model sizes on ImageNet $256 \times 256$. Our method consistently achieves lower FID and higher IS compared to MAR.

| Size | Method | FID ↓ | IS ↑ |
|---|---|---|---|
| 208M | MAR (Li et al., 2024a) | 2.31 | 281.7 |
| | **Ours** | **1.96** | **290.5** |
| 479M | MAR (Li et al., 2024a) | 1.78 | 296.0 |
| | **Ours** | **1.59** | **301.5** |
| 943M | MAR (Li et al., 2024a) | 1.55 | 303.7 |
| | **Ours** | **1.31** | **324.2** |

Table 3: Performance comparison on high-resolution ImageNet $512 \times 512$.

| Method | FID ↓ | IS ↑ |
|---|---|---|
| MAR (Li et al., 2024a) | 1.73 | 279.9 |
| **Ours** | **1.58** | **302.3** |

**High-Resolution Generation.**   To further assess the generalization capability of our method, we extended our evaluation to a higher resolution setting on ImageNet $512 \times 512$ (using a model size of approximately 481M parameters). Table 3 demonstrates the superiority of our approach in high-resolution synthesis. Our method achieves an FID of 1.58 compared to 1.73 for MAR, indicating that our OT-based condition refinement remains effective in mitigating inconsistencies even in higher-dimensional spaces.

## 5.4 CONDITION ERRORS ANALYSIS

Figure 3 presents the denoising analysis, showing Signal-to-Noise Ratio (SNR) and Noise Intensity over time for both our method and a baseline. The denoising process proceeds from right to left, with time steps decreasing as denoising progresses. Our method consistently achieves higher SNR, with a widening gap in later stages. Similarly, the right panel shows the Noise Intensity, where both methods show a reduction in noise as denoising progresses.

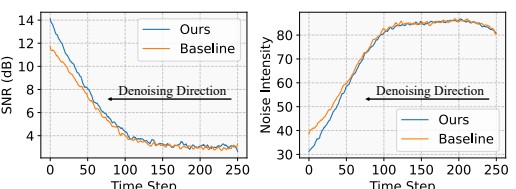

Figure 3: Analysis of Signal-to-Noise Ratio (SNR, **Left**) and Noise Intensity (**Right**) during the denoising process of our method and the baseline. All analyses are computed in the image space after VAE decoding.

Consistent with the SNR analysis, our method exhibits a marginally lower Noise Intensity, especially in the earlier time steps, i.e., later stages of denoising. These results highlight the efficacy of our proposed OT-based refinement of conditional distributions and effectively mitigate the potential inconsistencies introduced by purely autoregressive methods.

## 6 CONCLUSION

In this work, our analysis shows that patch denoising in autoregressive models mitigates condition errors, stabilizing condition distribution and enhancing generation quality. Autoregressive condition generation further refines conditions, exponentially reducing error influence. To address "condition inconsistency", we introduce a refinement method based on Optimal Transport and prove that casting it as Wasserstein Gradient Flow ensures convergence. Experimental results and analysis show the effectiveness of our method.

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

## A    RELATED WORK

### A.1    DIFFUSION MODEL

Diffusion models have emerged as a powerful generative framework, surpassing GANs (Goodfellow et al., 2020) and VAEs (Kingma & Welling, 2014) in stability and sample quality. DDPMs (Yang et al., 2024b) introduced a noise-based training and reconstruction paradigm, later linked theoretically to Score Matching and DAEs (Vincent, 2011). However, early diffusion models suffered from slow sampling due to numerous iterative steps. Improved DDPMs (Pang et al., 2024) refined noise scheduling, while DDIMs (Wei et al., 2024) accelerated generation through a non-Markovian formulation. LDMs (Rombach et al., 2022) further optimized efficiency by applying diffusion in a lower-dimensional latent space. Diffusion models also exhibit theoretical advantages over GANs, notably their implicit minimization of the Wasserstein distance (Kwon et al., 2022), leading to better convergence and robustness. Enhancing conditional control remains a key research focus: Classifier-Free Diffusion Guidance (Wahid et al., 2025) enables flexible conditioning without external classifiers, and structure-aware adaptations (Li & Yan, 2024) improve efficiency for structured data. Various applications extend their utility: Palette (Saharia et al., 2022) enhances image restoration, GLIDE (Nichol et al., 2022) improves text-guided synthesis, CDMs (Ho et al., 2022) refine images progressively, and ControlNet (Zhang et al., 2023) integrates structural conditions for enhanced controllability. Detecting diffusion-generated images is increasingly challenging, with studies like Schaefer et al. (Corvi et al., 2023) highlighting the need for robust detection methods.

### A.2    AUTOREGRESSIVE IMAGE GENERATION

Autoregressive models, despite their effectiveness, face computational constraints due to sequential generation (Zhou et al., 2024; Sun et al., 2025; Zhou et al., 2025b). Optimization efforts focus on efficiency and scalability: LlamaGen (Sun et al., 2024a) leverages large-scale training to surpass diffusion models in quality and efficiency, while VAR (Tian et al., 2024) reduces inference latency via next-scale prediction. Spatial alignment strategies like ImageFolder (Li et al., 2024b) improve autoregressive modeling, and Emu3 (Wang et al., 2024) unifies token prediction across modalities. Expanding autoregression to multimodal tasks requires bridging discrete and continuous data representations. Lumina-mGPT (Liu et al., 2024) employs a decoder-only Transformer for high-quality text-to-image synthesis, while MMAR (Yang et al., 2024a) models continuous tokens to enhance understanding and generation. Traditional vector quantization in autoregressive models is being reconsidered: VQ-free autoregression (Li et al., 2024a) introduces diffusion-based per-token probabilities for efficiency, and LatentLM (Sun et al., 2024b) integrates next-token diffusion for multimodal synthesis across image, speech, and text.

## B    LIMITATIONS

While our research provides novel theoretical insights and algorithmic advancements, it is important to acknowledge certain limitations. Specifically, our experimental evaluation has not been conducted on large-scale models due to the substantial computational resources required for such validation. Instead, our focus has been on a rigorous theoretical analysis and the development of scalable algorithms. Despite the absence of experiments on very large models, the generality and applicability of our method have been theoretically established, and our experiments on general settings support the soundness of the proposed approach. We believe future work can extend these evaluations to more resource-intensive settings to further verify empirical performance at scale.

## C    PROOF OF THEOREM 1 (CONDITIONAL SCORE MATCHING UPPER BOUND)

We begin by establishing two foundational lemmas required for the proof.

**Lemma 7** (Bayes' Theorem for Conditional Scores). *For any measurable sets $c$ and $\mathbf{x}_t$, the posterior distribution satisfies:*

$$p_t(c|\mathbf{x}_t) = \frac{p_c(c)p_t(\mathbf{x}_t|c)}{p_t(\mathbf{x}_t)} = \frac{p_c(c)p_t(\mathbf{x}_t|c)}{\int p_c(c')p_t(\mathbf{x}_t|c')dc'}$$

*where the second equality explicitly shows the marginalization over $c'$.*

**Lemma 8** (Jensen's Inequality for Convex Functions). *For any convex function $f : \mathbb{R}^d \to \mathbb{R}$ and random variable $\mathbf{Y}$ with finite expectation:*

$$f\left(\mathbb{E}[\mathbf{Y}]\right) \leq \mathbb{E}\left[f(\mathbf{Y})\right]$$

*Equality holds if and only if $f$ is affine linear on the support of $\mathbf{Y}$, or $\mathbf{Y}$ is constant almost surely.*

**Step-by-Step Proof:**   Using these lemmas, we proceed with the main proof.

**Step 1: Marginal-Conditional Decomposition.**   Express the marginal distribution through conditioning variables:

$$p_t(\mathbf{x}_t) = \int_{\mathcal{C}} p_c(c) p_t(\mathbf{x}_t|c) dc \quad \text{(Law of total probability)}$$

Differentiate both sides under the integral sign (valid under Dominated Convergence Theorem conditions):

$$\nabla_{\mathbf{x}_t} p_t(\mathbf{x}_t) = \int_{\mathcal{C}} p_c(c) \nabla_{\mathbf{x}_t} p_t(\mathbf{x}_t|c) dc$$

**Step 2: Score Function Representation.**   Using Lemma 7, decompose the marginal score:

$$\begin{aligned}
\nabla_{\mathbf{x}_t} \log p_t(\mathbf{x}_t) &= \frac{\nabla_{\mathbf{x}_t} p_t(\mathbf{x}_t)}{p_t(\mathbf{x}_t)} \\
&= \frac{\int p_c(c) \nabla_{\mathbf{x}_t} p_t(\mathbf{x}_t|c) dc}{p_t(\mathbf{x}_t)} \\
&= \int \underbrace{\frac{p_c(c) p_t(\mathbf{x}_t|c)}{p_t(\mathbf{x}_t)}}_{p_t(c|\mathbf{x}_t)} \nabla_{\mathbf{x}_t} \log p_t(\mathbf{x}_t|c) dc \\
&= \mathbb{E}_{c \sim p_t(c|\mathbf{x}_t)} \left[\nabla_{\mathbf{x}_t} \log p_t(\mathbf{x}_t|c)\right]
\end{aligned}$$

where the critical step (line 3) applies Lemma 7 to identify the posterior distribution.

**Step 3: Jensen's Inequality Application.**   Substitute into the unconditional loss:

$$\begin{aligned}
\mathbb{E}_{\mathbf{x}_t} \left\|\nabla \log p_t - s_\theta\right\|^2 &= \mathbb{E}_{\mathbf{x}_t} \left\|\mathbb{E}_{c|\mathbf{x}_t}[\nabla \log p_t(\cdot|c)] - s_\theta\right\|^2 \\
&\leq \mathbb{E}_{\mathbf{x}_t} \mathbb{E}_{c|\mathbf{x}_t} \left\|\nabla \log p_t(\cdot|c) - s_\theta\right\|^2 \quad \text{(by Lemma 8)}
\end{aligned}$$

Here we specifically apply Lemma 8 with:

- $f(\mathbf{y}) = \|\mathbf{y} - s_\theta\|^2$ (convex since $\|\cdot\|^2$ is convex)
- $\mathbf{Y} = \nabla_{\mathbf{x}_t} \log p_t(\mathbf{x}_t|c)$

**Step 4: Law of Total Expectation.**   Convert the nested expectation to a joint expectation:

$$\mathbb{E}_{\mathbf{x}_t} \mathbb{E}_{c|\mathbf{x}_t}[\cdot] = \mathbb{E}_{c,\mathbf{x}_t}[\cdot] = \mathbb{E}_{c \sim p_c} \mathbb{E}_{\mathbf{x}_t \sim p_t(\cdot|c)}[\cdot]$$

Thus, we obtain the final inequality:

$$\mathbb{E}_{\mathbf{x}_t} \left\|\nabla \log p_t - s_\theta\right\|^2 \leq \mathbb{E}_{c,\mathbf{x}_t} \left\|\nabla \log p_t(\cdot|c) - s_\theta\right\|^2$$

**Tightness Analysis.**   By Lemma 8, equality holds iff $\nabla_{\mathbf{x}_t} \log p_t(\mathbf{x}_t|c)$ is constant $c$-a.s., which requires $p_t(\mathbf{x}_t|c) = p_t(\mathbf{x}_t)$ for all $c$ in the support of $p_c$. This corresponds to statistical independence $\mathbf{x}_t \perp\!\!\!\perp c$.

## D   DETAILED DERIVATIONS FOR CONDITIONAL SCORE MATCHING ANALYSIS

### D.1   PROOF OF LEMMA 1

Following the score matching framework from (Vincent, 2011), we begin by expanding the squared norm in both unconditional and conditional score matching objectives.

**Unconditional Case:** For the unconditional score matching loss:

$$\mathbb{E}_{x_t \sim p_t(x_t)} \left[ \| \nabla_{x_t} \log p_t(x_t) - s_\theta(x_t, t) \|^2 \right]$$

$$= \mathbb{E}_{x_t \sim p_t(x_t)} \left[ \left( \nabla_{x_t} \log p_t(x_t) - s_\theta(x_t, t) \right)^\top \left( \nabla_{x_t} \log p_t(x_t) - s_\theta(x_t, t) \right) \right]$$

$$= \mathbb{E}_{x_t} \Big[ \underbrace{\| \nabla_{x_t} \log p_t(x_t) \|^2}_{\text{True score norm}} + \underbrace{\| s_\theta(x_t, t) \|^2}_{\text{Learned score norm}}$$

$$- 2 \underbrace{\langle s_\theta(x_t, t), \nabla_{x_t} \log p_t(x_t) \rangle}_{\text{Alignment term}} \Big]$$

This follows directly from the identity $\|a - b\|^2 = \|a\|^2 + \|b\|^2 - 2a^\top b$.

**Conditional Case:** For conditional score matching, expanding the L2 norm of the training error, we have:

$$E_{c, x_t \sim p_c(c) p_t(x_t|c)} \left[ \| \nabla_{x_t} \log p_t(x_t|c) - s_\theta(x_t, t) \|^2 \right]$$

$$= E_{c, x_t \sim p_c(c) p_t(x_t|c)} \left[ \| \nabla_{x_t} \log p_t(x_t|c) \|^2 + \| s_\theta(x_t, t) \|^2 - 2 \langle s_\theta(x_t, t), \nabla_{x_t} \log p_t(x_t|c) \rangle \right]$$

$$= E_{x_t \sim p_t(x_t)} E_{c \sim p_t(c|x_t)} \left[ \| \nabla_{x_t} \log p_t(x_t|c) \|^2 + \| s_\theta(x_t, t) \|^2 - 2 \langle s_\theta(x_t, t), \nabla_{x_t} \log p_t(x_t|c) \rangle \right]$$

$$= E_{x_t \sim p_t(x_t)} \left[ E_{c \sim p_t(c|x_t)} \left[ \| \nabla_{x_t} \log p_t(x_t|c) \|^2 \right] + \| s_\theta(x_t, t) \|^2 - 2 E_{c \sim p_t(c|x_t)} \langle s_\theta(x_t, t), \nabla_{x_t} \log p_t(x_t|c) \rangle \right]$$

## D.2 Derivation of Error Difference $\epsilon_c$ (Definition 1)

**Loss Difference Analysis:** Subtracting the unconditional loss from the conditional loss (after expansion):

$$\mathbb{E}_{c, x_t}[\text{Loss}] - \mathbb{E}_{x_t}[\text{Loss}]$$

$$= \mathbb{E}_{x_t} \mathbb{E}_{c|x_t} \left[ \| \nabla_{x_t} \log p_t(x_t|c) \|^2 \right] - \mathbb{E}_{x_t} \| \nabla_{x_t} \log p_t(x_t) \|^2$$

$$+ \underbrace{\mathbb{E}_{x_t} \| s_\theta \|^2 - \mathbb{E}_{x_t} \| s_\theta \|^2}_{=0}$$

$$- 2 \underbrace{\left( \mathbb{E}_{x_t, c|x_t} \langle s_\theta, \nabla \log p_t(x_t|c) \rangle - \mathbb{E}_{x_t} \langle s_\theta, \nabla \log p_t(x_t) \rangle \right)}_{\text{Vanishes by tower property}}$$

The cross-terms cancel due to the tower property of expectation:

$$\mathbb{E}_{x_t} \mathbb{E}_{c|x_t} \langle s_\theta, \nabla \log p_t(x_t|c) \rangle = \mathbb{E}_{x_t} \langle s_\theta, \mathbb{E}_{c|x_t} \nabla \log p_t(x_t|c) \rangle$$

$$= \mathbb{E}_{x_t} \langle s_\theta, \nabla \log p_t(x_t) \rangle$$

where we used the identity $\nabla \log p_t(x_t) = \mathbb{E}_{c|x_t} \nabla \log p_t(x_t|c)$ from (Li & Yan, 2024).

**Final Error Expression:** Thus, the difference reduces to:

$$\epsilon_c = \frac{1}{T} \sum_{t=1}^{T} \mathbb{E}_{x_t} \left[ \mathbb{E}_{c|x_t} \| \nabla \log p_t(x_t|c) \|^2 - \| \nabla \log p_t(x_t) \|^2 \right]$$

This quantifies the excess "score energy" induced by conditioning, similar to variance decomposition in probability theory.

## D.3 Properties of $\bar{\epsilon}_c$ (Definition 2)

From Definition 2, we can relate $\bar{\epsilon}_c$ to $\epsilon_c$ using the law of total variance:

$$\bar{\epsilon}_c = \epsilon_c + \frac{1}{T} \sum_{t=1}^{T} \mathbb{E}_{x_t} \| \nabla \log p_t(x_t) \|^2$$

This decomposition reveals that $\bar{\epsilon}_c$ contains both the intrinsic score energy from the unconditional distribution and the additional energy $\epsilon_c$ from conditioning.

## E  CONDITIONAL CONTROL TERM UNIQUENESS PROOF

*Proof.* We prove Lemma 2 through three key steps:

**Step 1: Bayesian Score Decomposition**    Using Bayes' rule $p_t(x_t|c) = \frac{p(c|x_t)p_t(x_t)}{p(c)}$, we derive:

$$\nabla_{x_t} \log p_t(x_t|c) = \nabla_{x_t} \log p(c|x_t) + \nabla_{x_t} \log p_t(x_t) - \underbrace{\nabla_{x_t} \log p(c)}_{=0}$$

$$= \nabla_{x_t} \log p(c|x_t) + \nabla_{x_t} \log p_t(x_t)$$

**Step 2: Cross-Term Cancellation**    The squared norm decomposes as:

$$\|\nabla_{x_t} \log p_t(x_t|c)\|^2 = \|\nabla_{x_t} \log p(c|x_t)\|^2 + \|\nabla_{x_t} \log p_t(x_t)\|^2$$
$$+ 2\langle \nabla_{x_t} \log p(c|x_t), \nabla_{x_t} \log p_t(x_t)\rangle$$

Taking expectation over $p_t(x_t)$ and $p_c(c)$:

$$\mathbb{E}[\langle \nabla_{x_t} \log p(c|x_t), \nabla_{x_t} \log p_t(x_t)\rangle] = \mathbb{E}_{x_t}\left[\mathbb{E}_c\left[\langle \nabla_{x_t} \log p(c|x_t), \nabla_{x_t} \log p_t(x_t)\rangle \mid x_t\right]\right]$$

$$= \mathbb{E}_{x_t}\left[\langle \underbrace{\mathbb{E}_c[\nabla_{x_t} \log p(c|x_t)]}_{=0}, \nabla_{x_t} \log p_t(x_t)\rangle\right] = 0$$

where the inner expectation vanishes because $\mathbb{E}_c[\nabla_{x_t} \log p(c|x_t)] = \nabla_{x_t} \mathbb{E}_c[p(c|x_t)] = \nabla_{x_t} 1 = 0$.

**Step 3: Variance Propagation**    Combining results from Steps 1-2:

$$\epsilon_c = \mathbb{E}\|\nabla_{x_t} \log p_t(x_t|c)\|^2 - \mathbb{E}\|\nabla_{x_t} \log p_t(x_t)\|^2$$
$$= \mathbb{E}\|\sigma_t^2 \nabla_{x_t} \log p(c|x_t)\|^2$$

The scaling factor $\sigma_t^2$ emerges from the reverse process parameterization in (Ho & Salimans, 2022; Liu et al., 2023), where the conditional mean adjustment contains an explicit $\sigma_t^2$ multiplier. This completes the proof that $\epsilon_c$ isolates the conditional control term's contribution. □

## F  PROOF OF CONDITION REFINEMENT VIA PATCH DENOISING (PROPOSITION 1)

To understand the long-term behavior of the condition refinement process, we invoke the Markov Chain Stationary Theorem. The conditional probability density function $p(x|c)$ is then given by: aBy invoking (Meyn & Tweedie, 2012), we establish that as $c_i$ iterates, its distribution converges to a stationary distribution. (The details of the lemma we used can be found in Appendix K.) Consequently, our analysis shifts to understanding how the gradient of the conditional probability evolves when $c_i$ follows a normal distribution. To compute the gradient of the conditional probability density function $p(x|c)$ with respect to $x$, the joint distribution of $x$ and $c$ follows a multivariate normal distribution:

$$\begin{pmatrix} X \\ C \end{pmatrix} \sim \mathcal{N}\left(\begin{pmatrix} \mu_x \\ \mu_c \end{pmatrix}, \begin{pmatrix} \sigma_{xx} & \sigma_{xc} \\ \sigma_{xc} & \sigma_{cc} \end{pmatrix}\right).$$

The conditional probability density function $p(x|c)$ is:

$$p(x|c) = \frac{1}{\sqrt{2\pi(\sigma_{xx} - \frac{\sigma_{xc}^2}{\sigma_{cc}})}} \exp\left(-\frac{(x - (\mu_x + \frac{\sigma_{xc}}{\sigma_{cc}}(c - \mu_c)))^2}{2(\sigma_{xx} - \frac{\sigma_{xc}^2}{\sigma_{cc}})}\right)$$

Taking the logarithm, we obtain the log-likelihood function:

$$\log p(x|c) = -\frac{1}{2}\log\left(2\pi(\sigma_{xx} - \frac{\sigma_{xc}^2}{\sigma_{cc}})\right)$$
$$-\frac{(x - (\mu_x + \frac{\sigma_{xc}}{\sigma_{cc}}(c - \mu_c)))^2}{2(\sigma_{xx} - \frac{\sigma_{xc}^2}{\sigma_{cc}})}$$

Differentiating with respect to $x$, while ignoring constant terms, yields:

$$\nabla_x \log p(x|c) = -\frac{x - (\mu_x + \frac{\sigma_{xc}}{\sigma_{cc}}(c - \mu_c))}{\sigma_{xx} - \frac{\sigma_{xc}^2}{\sigma_{cc}}}.$$

The squared norm of this gradient is given by:

$$\|\nabla_x \log p(x|c)\|^2 = \left(\frac{x - (\mu_x + \frac{\sigma_{xc}}{\sigma_{cc}}(c - \mu_c))}{\sigma_{xx} - \frac{\sigma_{xc}^2}{\sigma_{cc}}}\right)^2.$$

The conditional mean and the conditional variance primarily influence the gradient behavior. The conditional mean is:

$$\mu_x + \frac{\sigma_{xc}}{\sigma_{cc}}(c - \mu_c),$$

where $\mu_c$ represents the mean of $c$. As $c$ iterates and reaches its stationary distribution, $\mu_c$ converges to a constant, which we denote as $\mu_c^{\text{stable}}$. Consequently, the conditional mean stabilizes to a fixed value. The conditional variance $\sigma_{xx} - \frac{\sigma_{xc}^2}{\sigma_{cc}}$ is determined by the covariance structure of the joint distribution. Since this variance does not depend on $c$ in its stationary distribution, it remains unchanged. As $c$ reaches its stationary distribution, the deviation of $x$ from the conditional mean gradually diminishes while the variance remains constant. It leads to gradient magnitude decay, indicating attenuation of the conditional probability gradient over iterations.

## G    PROOF OF DESCENT OF GRADIENT NORM IN AUTOREGRESSIVE PROCESS (THEOREM 2)

*Proof.*

**Remark 2** (State Space Representation). By introducing the state vector $\mathbf{c}_i = (c_i, c_{i-1}, \ldots, c_{i-p+1})^\top$, we can represent the original process as a vector-valued first-order autoregressive process:

$$\mathbf{c}_{i+1} = \mathbf{A}\mathbf{c}_i + \mathbf{e}_{i+1}$$

$$\mathbf{A} = \begin{pmatrix} a_0 & a_1 & \cdots & a_{p-2} & a_{p-1} \\ 1 & 0 & \cdots & 0 & 0 \\ 0 & 1 & \cdots & 0 & 0 \\ \vdots & \vdots & \ddots & \vdots & \vdots \\ 0 & 0 & \cdots & 1 & 0 \end{pmatrix}, \quad \mathbf{e}_{i+1} = \begin{pmatrix} \varepsilon_{i+1} \\ 0 \\ 0 \\ \vdots \\ 0 \end{pmatrix}$$

In this representation, the conditional distribution of $\mathbf{c}_{i+1}$ depends only on $\mathbf{c}_i$. From Assumption 4 (1), the spectral radius of matrix $\mathbf{A}$ is less than 1, which ensures the stability of the process.

Let $p_t(x_t|\pi) = \int_{\mathcal{X}} p_t(x_t|c)\pi(dc)$. The log-likelihood gradient can be decomposed as:

$$\begin{aligned} \nabla_{x_t} \log p_t(x_t|c_i) &= \frac{\nabla_{x_t} p_t(x_t|c_i)}{p_t(x_t|c_i)} \\ &= \left(\frac{\nabla_{x_t} p_t(x_t|c_i)}{p_t(x_t|c_i)} - \frac{\nabla_{x_t} p_t(x_t|\pi)}{p_t(x_t|\pi)}\right) + \underbrace{\frac{\nabla_{x_t} p_t(x_t|\pi)}{p_t(x_t|\pi)}}_{\text{Stationary Term}} \end{aligned}$$

Using the triangle inequality for norms, we get:

$$\|\nabla_{x_t} \log p_t(x_t|c_i)\| \leq \left\|\frac{\nabla_{x_t} p_t(x_t|c_i)}{p_t(x_t|c_i)} - \frac{\nabla_{x_t} p_t(x_t|\pi)}{p_t(x_t|\pi)}\right\| + \left\|\underbrace{\frac{\nabla_{x_t} p_t(x_t|\pi)}{p_t(x_t|\pi)}}_{\text{Stationary Term}}\right\|$$

For the non-stationary term, we have:

$$\left\| \frac{\nabla_{x_t} p_t(x_t|c_i)}{p_t(x_t|c_i)} - \frac{\nabla_{x_t} p_t(x_t|\pi)}{p_t(x_t|\pi)} \right\|$$

$$= \left\| \frac{p_t(x_t|\pi)\nabla_{x_t} p_t(x_t|c_i) - p_t(x_t|c_i)\nabla_{x_t} p_t(x_t|\pi)}{p_t(x_t|c_i)p_t(x_t|\pi)} \right\|$$

$$\leq \frac{1}{\delta^2} \left\| p_t(x_t|\pi)\nabla_{x_t} p_t(x_t|c_i) - p_t(x_t|c_i)\nabla_{x_t} p_t(x_t|\pi) \right\|$$

$$= \frac{1}{\delta^2} \left\| p_t(x_t|\pi)[\nabla_{x_t} p_t(x_t|c_i) - \nabla_{x_t} p_t(x_t|\pi)] + \nabla_{x_t} p_t(x_t|\pi)[p_t(x_t|\pi) - p_t(x_t|c_i)] \right\|$$

$$\leq \frac{1}{\delta^2} \Big[ \left\| p_t(x_t|\pi)[\nabla_{x_t} p_t(x_t|c_i) - \nabla_{x_t} p_t(x_t|\pi)] \right\| + \left\| \nabla_{x_t} p_t(x_t|\pi)[p_t(x_t|\pi) - p_t(x_t|c_i)] \right\| \Big]$$

Using the Lipschitz property of both the conditional probability density function $p_t(x_t|c)$ and its gradient $\nabla_{x_t} p_t(x_t|c)$ with respect to $c$ (with Lipschitz constant $L$), and the upper bound $M_2$ for $p_t(x_t|\pi)$ from Lemma 4, the above inequality becomes:

$$\leq \frac{1}{\delta^2} \Big[ M_2 L \|c_i - c_\pi\| + \|\nabla_{x_t} p_t(x_t|\pi)\| L \|c_i - c_\pi\| \Big]$$

$$= \frac{L}{\delta^2} \Big[ M_2 + \|\nabla_{x_t} p_t(x_t|\pi)\| \Big] \|c_i - c_\pi\|$$

We consider $c_\pi$ to be a representative value from the stationary distribution $\pi$, for simplicity we can consider $c_\pi = \mathbb{E}_\pi[c]$. Applying the geometric ergodicity of the Markov chain (Lemma 9 in Appendix H), which gives us $\|c_i - c_\pi\| \leq \sqrt{\mathrm{Var}_\pi(c)}\rho^i$, we arrive at:

$$\leq \frac{L}{\delta^2} \Big[ M_2 + \|\nabla_{x_t} p_t(x_t|\pi)\| \Big] \sqrt{\mathrm{Var}_\pi(c)}\rho^i$$

For the stationary term, from Lemma 5, we have:

$$\left\| \frac{\nabla_{x_t} p_t(x_t|\pi)}{p_t(x_t|\pi)} \right\| \leq \frac{\|\nabla_{x_t} p_t(\cdot|\pi)\|_\infty}{\delta} \leq \frac{M_1}{\delta} := m$$

where $m$ is a constant.

Combining the estimates for both terms, we have:

$$\|\nabla_{x_t} \log p_t(x_t|c_i)\| \leq \frac{L}{\delta^2}\Big(M_2 + \|\nabla_{x_t} p_t(x_t|\pi)\|\Big)\sqrt{\mathrm{Var}_\pi(c)}\rho^i + \frac{M_1}{\delta}$$

$$\leq M\rho^i + m$$

where $M = \frac{L}{\delta^2}\Big(M_2 + \|\nabla_{x_t} p_t(x_t|\pi)\|\Big)\sqrt{\mathrm{Var}_\pi(c)}$ and $\beta = \rho \in (0, 1)$.

Thus, we obtain the desired exponential decay estimate for the gradient norm as the autoregressive process iterates. This estimate holds uniformly for all $x_t \in \mathcal{X}$. $\qquad\square$

## H  GEOMETRIC ERGODICITY AND CONVERGENCE TO STATIONARY MEAN

**Lemma 9** (Geometric Ergodicity and Convergence to Stationary Mean). *Assume that the Markov chain $\{c_t\}_{t\geq 0}$ is geometrically ergodic with stationary distribution $\pi$, and let $c_\pi = \mathbb{E}_\pi[c]$. Then, there exist constants $C > 0$ and $0 < \rho < 1$ such that for all $i \geq 0$:*

$$\|c_i - c_\pi\| \leq C\rho^i \sqrt{\mathrm{Var}_\pi(c)}$$

*where $\mathrm{Var}_\pi(c) = \mathbb{E}_\pi[\|c - c_\pi\|^2]$.*

*Explanation:* Lemma 9 formalizes the geometric convergence of the Markov chain state $c_i$ towards the stationary mean $c_\pi$ due to geometric ergodicity. This property ensures that the influence of the initial condition diminishes exponentially over time, allowing us to bound the distance $\|c_i - c_\pi\|$ by a geometrically decaying term proportional to $\sqrt{\mathrm{Var}_\pi(c)}$. This justifies the transition in the eighth line of the derivation, replacing $\|c_i - c_\pi\|$ with a term that decays geometrically with $i$.

## I    PROOF OF REGULARITY OF CONDITIONAL PROBABILITIES

1. Existence of a lower bound: Since the conditional probability density is quadratically continuous and differentiable and is defined on a tight set, there is a minimum by the extreme value theorem. Since the probability density is always positive, the minimum must be greater than zero.

2. Using the boundedness of the second-order derivatives in our Assumption 2

$$for \quad \forall x \quad \exists K > 0, \quad \|\nabla^2_{x_t} p_t(x_t|c_i)\| \leq K$$

Consider the difference of derivatives:

$$\nabla_{x_t} p_t(x_t|c_1) - \nabla_{x_t} p_t(x_t|c_2)$$

By the median theorem in multivariable calculus, there exists some point between $c_1 c_2$ such that:

$$\nabla_{x_t} p_t(x_t|c_1) - \nabla_{x_t} p_t(x_t|c_2) = \nabla^2_{x_t} p_t(x_t|c)(c_1 - c_2)$$

By taking the value of the paradigm of the above equation, we have:

$$\|\nabla_{x_t} p_t(x_t|c_1) - \nabla_{x_t} p_t(x_t|c_2)\| = \|\nabla^2_{x_t} p_t(x_t|c)(c_1 - c_2)\|$$

Using Multiplicative Inequality for Norms($\|a \cdot b\| \leq \|a\| \cdot \|b\|$)and the boundedness of the second-order derivatives, we obtain:

$$\|\nabla_{x_t} p_t(x_t|c_1) - \nabla_{x_t} p_t(x_t|c_2)\| \leq \|\nabla^2_{x_t} p_t(x_t|c)\| \cdot \|(c_1 - c_2)\| \leq K \cdot \|(c_1 - c_2)\|$$

Thus, taking L = M, we get the final inequality:

$$\|\nabla_{x_t} p_t(x_t|c_1) - \nabla_{x_t} p_t(x_t|c_2)\| \leq L\|(c_1 - c_2)\|$$

## J    PROOF OF THE MARKOV PROPERTY

This paper mainly uses the last geometric traversal of the theorem and therefore focuses on proving the geometric traversal of Markov chains. The proof is divided into three steps.

1. Drift Conditional Verification: Constructing Foster-Lyapunov Functions $V(x) = 1 + |x|^2$, for $\quad \forall x$

$$\begin{aligned} E[V(X_{n+1})|X_n = x] &= 1 + E[|a_n x + \epsilon_n|^2] \\ &= 1 + |a_n|^2|x|^2 + \sigma^2 \\ &= |a_n|^2(1 + |x|^2) + (1 - |a_n|^2 + \sigma^2) \\ &\leq \lambda V(x) + b \end{aligned}$$

where$\lambda = \sup_n |a_n|^2 < 1(\sum |a_i| < \infty)$, $b = 1 + \sigma^2$.

2. Compactness:for $\forall R > 0$, the set $\{x : V(x) \leq R\}$ is compact because it is equivalent to the closed ball$\{x : |x|^2 \leq R - 1\}$.

3. Irregularity and continuity: Since the noise term $\epsilon_n$ obeys a normal distribution, the transfer probability has a positive density everywhere, which guarantees strong Feller and irregularity of the chain.
   According to Meyn-Tweedie theory, the above condition guarantees geometric ergodicity.

## K    LEMMA OF MARKOV PROPERTY

**Lemma 10** (Markov Chain Stationary Theorem). *If a random process has a transition matrix $P$ and is ergodic (i.e., any two states are aperiodic and irreducible), then:*

1. *The limit of the $n$-step transition matrix exists and is given by:*

$$\lim_{n \to \infty} P^n = \begin{pmatrix} \pi(1) & \pi(2) & \cdots & \pi(j) & \cdots \\ \pi(1) & \pi(2) & \cdots & \pi(j) & \cdots \\ \vdots & \vdots & \ddots & \vdots & \ddots \\ \pi(1) & \pi(2) & \cdots & \pi(j) & \cdots \\ \vdots & \vdots & \vdots & \vdots & \ddots \end{pmatrix}$$

2. *The stationary distribution $\pi = [\pi(1), \pi(2), \ldots]$ satisfies the equation:*

$$\pi(j) = \sum_i \pi_i P_{ij}$$

3. $\pi$ *is the unique non-negative solution to the stationary equation, with $\sum_i \pi(i) = 1$.*

## L  AUTOREGRESSIVE CONDITION OPTIMIZATION ALGORITHM

The full algorithm integrates three key components: (1) autoregressive condition generation, (2) diffusion-based denoising, and (3) optimal transport refinement. The pseudocode below specifies the detailed computational workflow.

---

**Algorithm 1** Autoregressive Condition Optimization (ACO) with Denoising Integration

---

**Require:** Initial condition $c_0 \leftarrow \Phi_\theta(c_{i-1}, x_{<i})$; Diffusion model $\{\mathcal{D}_t\}_{t=1}^T$ with noise levels $\{\beta_t\}$; Target latent distribution $P_{z^*}$, OT parameters $\lambda, \epsilon, K_{\text{sink}}$; Learning rate schedule $\{\eta_k\}$, gradient clipping threshold $\tau$

**Ensure:** Optimized condition $c_i^*$, generated latent $z_i^{(T)}$

1: **Initialize:** $c^{(0)} \leftarrow c_0$, $z^{(0)} \sim \mathcal{N}(0, I)$
2: **for** $k = 0$ **to** $K - 1$ **do**
3:     **Denoising trajectory:**
4:     **for** $t = T$ **to** $1$ **do**
5:         $z^{(k,t-1)} \leftarrow \mathcal{D}_t(z^{(k,t)}, c^{(k)})$ {DDIM update}
6:     **end for**
7:     **Inverse process alignment:**
8:     $\phi(c^{(k)}) = \|c^{(k)} - \mathcal{T}^{-1}(z^{(k,0)})\|^2 + \alpha\|\nabla_z \mathcal{T}^{-1}\|_F^2$
9:     **Optimal transport computation:**
10:     Sample reference latents $\{z_j^*\} \sim P_{z^*}$
11:     Compute pairwise cost matrix: $C_{mn} = \underbrace{\|z_m^{(k,0)} - z_n^*\|^2}_{\text{Latent matching}} + \lambda \underbrace{\|c_m^{(k)} - \mathcal{T}^{-1}(z_n^*)\|^2}_{\text{Condition consistency}}$
12:     **Entropy-regularized OT:**
13:     Initialize $u^{(0)} \leftarrow \mathbf{1}$, $v^{(0)} \leftarrow \mathbf{1}$
14:     **for** $l = 1$ **to** $K_{\text{sink}}$ **do**
15:         $v^{(l)} \leftarrow \frac{P_{z^*}}{K_\epsilon(u^{(l-1)}, v^{(l-1)})}$
16:         $u^{(l)} \leftarrow \frac{P_z^{(k)}}{K_\epsilon(u^{(l-1)}, v^{(l)})}$
17:     **end for**
18:     $\gamma^{(k)} \leftarrow \text{diag}(u^{(K_{\text{sink}})}) \cdot K_\epsilon \cdot \text{diag}(v^{(K_{\text{sink}})})$
19:     **Gradient computation & update:**
20:     $\nabla_c \mathcal{L}_{\text{OT}} \leftarrow \gamma^{(k)} \odot \frac{\partial C}{\partial c^{(k)}}$
21:     $\nabla_c \mathcal{L}_{\text{reg}} \leftarrow \frac{\partial \phi}{\partial c^{(k)}}$
22:     $\nabla_c^{\text{total}} \leftarrow \text{Clip}(\nabla_c \mathcal{L}_{\text{OT}} + \nabla_c \mathcal{L}_{\text{reg}}, \tau)$
23:     $c^{(k+1)} \leftarrow c^{(k)} - \eta_k \nabla_c^{\text{total}}$
24: **end for**
25: **Return** $c_i^* \leftarrow c^{(K)}$, $z_i \leftarrow z^{(K,0)}$

---

### L.1 IMPLEMENTATION DETAILS

- **Target Distribution Estimation**: Maintain an EMA of generated latents:

$$P_{z^*}^{(i)} = (1 - \nu)P_{z^*}^{(i-1)} + \nu \frac{1}{B} \sum_{b=1}^{B} \delta(z_b^{(i)})$$

  with $\nu = 0.1$ and buffer size $B = 2048$.
- **Adaptive Entropy Regularization**: Schedule $\epsilon$ during Sinkhorn iterations:

$$\epsilon^{(k)} = \epsilon_{\max} - (\epsilon_{\max} - \epsilon_{\min})\frac{k}{K}$$

- **Stochastic Optimization**: Use Adam optimizer with:

$$\eta_k = \eta_0 \cdot \min(1, \sqrt{k_{\mathrm{warm}}/k})$$

  where $k_{\mathrm{warm}} = 100$ controls learning rate warmup.

### L.2 CONVERGENCE ANALYSIS

The algorithm maintains the convergence guarantee in Theorem 3 through:

1. **Monotonic Improvement**: For Lyapunov function

$$\mathcal{V}_k = W_2(P_{z^{(k)}}, P_{z^*}) + \lambda_{\mathrm{reg}}\mathbb{E}[\phi(c^{(k)})]$$

   we have $\mathcal{V}_{k+1} \leq \mathcal{V}_k - \eta_k \|\nabla \mathcal{V}_k\|^2 + \mathcal{O}(\eta_k^2)$
2. **Error Propagation Bound**: Approximation error from Sinkhorn iterations satisfies

$$\|\gamma^{(k)} - \gamma^*\|_F \leq C\rho^{K_{\mathrm{sink}}}$$

   with $\rho = \frac{\epsilon}{\epsilon+\delta}$ where $\delta$ is the minimum cost matrix entry.
3. **Stability Condition**: Gradient clipping ensures Lipschitz continuity:

$$\|\nabla_c^{\mathrm{total}}\|_2 \leq \tau(1 + \lambda_{\mathrm{reg}}L_{\mathcal{T}^{-1}})$$

   where $L_{\mathcal{T}^{-1}}$ is the Lipschitz constant of the inverse process.

## M  BOUNDED THEOREM

Function $p_t(x_t|c)$ in the fixed interval $[a, b]$ has second order derivatives $\nabla_{x_t}^2 p_t(x_t|c)$, and $\nabla_{x_t}^2 p_t(x_t|c)$ is bounded, so we have $K > 0$,

$$|\nabla_{x_t}^2 p_t(x_t|c)| \leq K, \quad \forall x_t \in [a, b].$$

### THE FIRST DERIVATIVE IS BOUNDED

We use the Mean Value Theorem to show that first-order derivatives are bounded. According to the mean value theorem, if $x_t, x_t' \in [a, b]$, there exists a point $\xi \in (x_t, x_t')$, then:

$$\nabla_{x_t} p_t(x_t'|c) - \nabla_{x_t} p_t(x_t|c) = \nabla_{x_t}^2 p_t(\xi|c)(x_t' - x_t).$$

$|\nabla_{x_t}^2 p_t(x_t|c)| \leq C$, so we have:

$$|\nabla_{x_t} p_t(x_t'|c) - p_t'(x_t|c)| = |\nabla_{x_t}^2 p_t(\xi|c)(x_t' - x_t)| \leq C|x_t' - x_t|.$$

This shows that $\nabla_{x_t} p_t(x_t|c)$ is Lipschitz continuous and the Lipschitz constant is $C$. So $\nabla_{x_t} p_t(x_t|c)$ is bounded. Next, we give specific boundedness estimates Taking $x_t = a$, we have:

$$|\nabla_{x_t} p_t(x_t|c) - \nabla_{x_t} p_t(a|c)| \leq C|x_t - a|.$$

Since $|x_t - a| \leq b - a$, we get:

$$|\nabla_{x_t} p_t(x_t|c)| \leq |\nabla_{x_t} p_t(a|c)| + C(b - a).$$

Therefore, there exists a constant $M_1 = |\nabla_{x_t} p_t(a|c)| + C(b - a)$ such that for all $x_t \in [a, b]$:

$$|\nabla_{x_t} p_t(x_t|c)| \leq M_1.$$

THE ORIGINAL FUNCTION IS BOUNDED

According to the Fundamental Theorem of Calculus, we have:

$$p_t(x_t|c) - p_t(a|c) = \int_a^{x_t} \nabla_{x_t} p_t(y|c) \, dy.$$

Since $|\nabla_{x_t} p_t(x_t|c)| \leq M_1$ for all $x_t \in [a, b]$, we can estimate the above integral:

$$|p_t(x_t|c) - p_t(a|c)| = \left| \int_a^{x_t} \nabla_{x_t} p_t(y|c) \, dy \right| \leq \int_a^{x_t} |\nabla_{x_t} p_t(y|c)| \, dy \leq M_1 |x_t - a|.$$

Since $|x_t - a| \leq b - a$, we have:

$$|p_t(x_t|c) - p_t(a|c)| \leq M_1(b - a).$$

Therefore, the original function $p_t(x_t|c)$ is bounded and:

$$|p_t(x_t|c)| \leq |p_t(a|c)| + M_1(b - a).$$

Thus, there exists the constant $M_2 = |p_t(a|c)| + M_1(b - a)$ such that for all $x_t \in [a, b]$:

$$|p_t(x_t|c)| \leq M_2.$$

Thus, in a fixed interval, a bounded second-order derivative is bounded by a bounded first-order derivative, and the original function is bounded by the proof.

## N   THE USE OF LARGE LANGUAGE MODELS

In this paper, ChatGPT was employed to assist in polishing the writing. The model was used as a language aid to improve clarity, grammar, and readability of the text, while ensuring that the academic content and arguments remain entirely the work of the author.

## O   TABLE OF NOTATIONS

To facilitate easier reading of the theoretical sections and provide a quick reference for the mathematical symbols used throughout the paper, we summarize the key notations in Table 4.

Table 4: Summary of Notations

| Symbol | Description |
|---|---|
| *Diffusion & Autoregressive Basics* | |
| $x_0, x_T$ | The original data (image) and the Gaussian noise at time $T$. |
| $x_{1:T}$ | The sequence of latent variables in the forward diffusion process. |
| $q(x_t\|x_{t-1})$ | The forward diffusion transition kernel. |
| $p_\theta(x_{t-1}\|x_t)$ | The reverse (denoising) process approximated by the network. |
| $s_\theta(x_t, t)$ | The score function predicted by the neural network. |
| $c$ | A global, static condition (in standard conditional diffusion). |
| $c_i$ | The autoregressively generated condition for the $i$-th patch. |
| $x_i$ | The $i$-th image patch. |
| $x_{<i}$ | The set of patches preceding $i$, i.e., $\{x_1, \ldots, x_{i-1}\}$. |
| $\Phi_\theta, \Gamma_\theta$ | The transition operator and noise modulation in the AR condition process. |
| *Error Analysis & Theory* | |
| $\epsilon_c$ | **Conditional Error Term** (Eq. 13). Measures the change in expected score squared norm due to conditioning. |
| $\bar{\epsilon}_c$ | **Simplified Conditional Error Term** (Eq. 14). Directly measures the expected squared norm of the conditional score. |
| $\mathcal{T}(c_i)$ | The transition function representing patch-based condition refinement. |
| $\sigma_t^2 \nabla_{x_t} \log p(c\|x_t)$ | The conditional guidance term in the reverse process mean. |
| $M, \beta, m$ | Constants and decay rate describing the descent of the gradient norm (Theorem 2). |
| *Condition Refinement (Optimal Transport)* | |
| $c_i^*$ | The ideal condition for patch $x_i$. |
| $\mathcal{I}_i^*$ | The minimal sufficient information subspace for patch $x_i$. |
| $\pi_{\mathcal{I}_i^*}$ | Orthogonal projection onto the subspace $\mathcal{I}_i^*$. |
| $\eta_i$ | **Extraneous Information Component** (Eq. 25). The deviation from the ideal condition ($\eta_i = c_i - c_i^*$). |
| $W_2(\cdot, \cdot)$ | The 2-Wasserstein distance between probability distributions. |
| $\mathcal{F}(P)$ | The free energy functional minimized by the Wasserstein Gradient Flow. |
| $T^{-1}$ | The inverse process regularization operator. |
| $P_c^{(k)}$ | The probability distribution of the condition at refinement step $k$. |
| $\rho$ | The contraction rate of the Wasserstein Gradient Flow (Theorem 3). |

