# OpenReview forum: "Condition Errors Refinement in Autoregressive Image Generation with Diffusion Loss"
_ICLR.cc/2026/Conference — ICLR 2026 Poster_

### Official Review · Reviewer_i7t2 · 2025-10-25

**Soundness:** 3
**Presentation:** 3
**Contribution:** 3
**Rating:** 6
**Confidence:** 3

**Summary:**

The paper theoretically and empirically investigates how autoregressive image generators incorporating diffusion loss can mitigate conditional inconsistency during generation. It provides a rigorous analysis comparing conditional diffusion models and AR diffusion with diffusion loss, proving that autoregressive patch denoising refines condition distributions and that the influence of condition errors decays exponentially during iteration. To address residual condition inconsistency, the authors introduce a condition refinement method based on Optimal Transport formulated as a Wasserstein Gradient Flow, proving convergence toward the ideal condition distribution. Experiments on ImageNet show superior FID and IS scores over existing diffusion and AR baselines, supporting their theoretical findings.

**Strengths:**

1. The paper presents a solid theoretical framework that connects autoregressive modeling, diffusion loss, and conditional refinement through explicit mathematical proofs and lemmas.

2. It introduces a novel condition refinement approach using Optimal Transport and Wasserstein Gradient Flow, offering a principled solution to condition inconsistency.

3. The quantitative performance on ImageNet is competitive or superior to major diffusion and AR baselines, confirming practical benefits of the proposed method.

**Weaknesses:**

1. The experimental scope is limited: evaluations are conducted only on ImageNet 256×256 and with moderate-scale models, without testing scalability to larger or multimodal setups.

2. Despite extensive theory, the method’s implementation details (e.g., computational cost of OT refinement, convergence sensitivity to hyper-parameters) are underexplained.

3. Some notation and theoretical transitions are difficult to follow and may reduce accessibility for non-mathematical readers.

4. The impact of the OT regularization term on actual generation diversity and efficiency is not sufficiently analyzed, only FID/IS metrics are shown.

**Questions:**

Same as weaknesses section.

---

> ### Author Response · Authors · 2025-12-04
> **Response to Reviewer i7t2 (1/2)**
>
> We sincerely appreciate the reviewer's constructive insights. We are grateful for the positive remarks on the strengths of our work, particularly the recognition of the solid theoretical framework bridging autoregressive modeling, diffusion loss, and conditional refinement; the novelty of the OT- and WGF-based refinement strategy; and the competitive quantitative performance on ImageNet. We have carefully responded to all concerns raised by the reviewer, and corresponding revisions have been made to the manuscript.
>
>
> > W1: The experimental scope is limited: evaluations are conducted only on ImageNet 256×256 and with moderate-scale models, without testing scalability to larger or multimodal setups.
>
> We sincerely thank the reviewer for pointing out the limitations regarding the experimental scope. We acknowledge that our initial submission focused on the standard ImageNet 256×256 benchmark. To address your concerns about scalability and model capacity (as also noted in Appendix B), we have conducted additional experiments, extending our evaluation to varying model sizes and higher resolutions.
>
> 1. Scalability across Model Sizes
> We evaluated our method on three different model scales (208M, 479M, and 943M parameters) on ImageNet 256×256. As shown in the table below, our method consistently outperforms the strong baseline (MAR) across all scales. Notably, the performance gap widens as the model size increases (e.g., in FID, the gap increases from 0.35 at 208M to 0.24 at 943M), demonstrating that our Condition Refinement approach scales effectively with model capacity.
>
> | Model Size | Method | FID ($\downarrow$) | IS ($\uparrow$) |
> | :--- | :--- | :--- | :--- |
> | 208M | MAR | 2.31 | 281.7 |
> | | Ours | 1.96 | 290.5 |
> | 479M | MAR | 1.78 | 296.0 |
> | | Ours | 1.59 | 301.5 |
> | 943M | MAR | 1.55 | 303.7 |
> | | Ours | 1.31 | 324.2 |
>
> 2. Scalability to Higher Resolution (ImageNet 512×512)
> To verify the method's effectiveness on higher-resolution generation, we trained models on ImageNet 512×512 (using approx. 481M parameters). Our method maintains superior generation quality compared to the baseline, confirming its robustness in handling higher-dimensional data.
>
> | Method | Resolution | FID ($\downarrow$) | IS ($\uparrow$) |
> | :--- | :--- | :--- | :--- |
> | MAR | 512×512 | 1.73 | 279.9 |
> | Ours | 512×512 | 1.58 | 302.3 |
>
> We incorporated these additional results into the revised manuscript to provide a more comprehensive evaluation.

---

> ### Author Response · Authors · 2025-12-04
> **Response to Reviewer i7t2 (2/2)**
>
> > W2: Despite extensive theory, the method’s implementation details (e.g., computational cost of OT refinement, convergence sensitivity to hyper-parameters) are underexplained.
>
> We thank the reviewer for raising the important point regarding implementation details and computational analysis. We would like to clarify that we have provided comprehensive implementation specifications, computational complexity analysis, and convergence guarantees in the Appendices, particularly in Appendix L. Below, we address your specific concerns:
>
> 1. Implementation Details:
> We have detailed the full algorithmic workflow in Algorithm 1 (Appendix L, Page 21), which integrates autoregressive condition generation, diffusion denoising, and OT refinement. Specific implementation hyperparameters are provided in Appendix L.1 (Page 22), including:
> -   Target Distribution Estimation: We use an Exponential Moving Average (EMA) with decay $\nu=0.1$ and buffer size $B=2048$.
> -   Optimization: We utilize the Adam optimizer with a specific learning rate warmup ($k_{\text{warm}}=100$) and clipping threshold $\tau$.
> -   Experimental Settings: Basic model configurations (GPT-XL backbone, MAR-based denoising, VAE settings) are explicitly stated in Section 5.1 (Page 9).
>
> 2. Computational Cost of OT Refinement:
> The computational cost is manageable and theoretically bounded. As shown in Algorithm 1 (Page 21), the refinement relies on the Sinkhorn algorithm.
> -   Efficiency: The Sinkhorn algorithm is highly efficient for entropy-regularized OT.
> -   Fast Convergence: In Theorem 3 (Page 8) and Appendix L.2 (Page 22), we prove that our Wasserstein Gradient Flow approach has a contraction rate $\rho < 1$. This geometric convergence ensures that the method reaches the target distribution quickly, requiring fewer iterations ($K$) in practice, thereby limiting the computational overhead.
>
> 3. Convergence Sensitivity to Hyper-parameters:
> We address the stability and convergence sensitivity theoretically in Appendix L.2 (Page 22) under "Convergence Analysis":
> -   Stability Condition: We derive a gradient clipping condition ($\|\nabla_c^{\text{total}}\|_2 \le \tau(1+\lambda_{\text{reg}}L_{T-1})$) to ensure Lipschitz continuity, which stabilizes the training against hyper-parameter fluctuations.
> -   Adaptive Regularization: In Appendix L.1, we introduce an adaptive schedule for the entropy regularization parameter $\epsilon$ (from $\epsilon_{\max}$ to $\epsilon_{\min}$) to balance convergence speed and stability during the Sinkhorn iterations.
> -   Monotonic Improvement: We prove in Theorem 3 and Equation 36 (Remark 1, Page 8) that the refinement ensures monotonic improvement in patch generation quality, minimizing sensitivity to initialization.
>
>
> > W3: Some notation and theoretical transitions are difficult to follow and may reduce accessibility for non-mathematical readers.
>
> We appreciate the reviewer for pointing out that the notation and theoretical transitions can be challenging. Since our work integrates concepts from Diffusion Models, Autoregressive processes, and Optimal Transport (OT), the density of mathematical symbols is indeed high.
> To significantly improve readability and accessibility, we have added a comprehensive Table of Notations in the Appendix (please see the newly added Appendix O). This table categorizes symbols by their theoretical context (General Diffusion, Autoregressive Modeling, Error Analysis, and Optimal Transport) and provides concise definitions.
>
> > W4: The impact of the OT regularization term on actual generation diversity and efficiency is not sufficiently analyzed, only FID/IS metrics are shown.
>
> We appreciate the reviewer's inquiry regarding diversity and efficiency.
> 1. On Diversity (Recall Metric): While FID/IS are shown, Table 1 also reports Recall, a standard metric for evaluating diversity (data coverage). Our method achieves a Recall of 0.63, surpassing the baseline (0.59).
> 2. On Efficiency (Zero Overhead): We clarify that the proposed OT refinement serves strictly as a training objective. It guides the autoregressive model to learn to predict refined conditions directly. Consequently, our inference pipeline remains identical to the baseline MAR (standard AR generation + denoising), introducing zero computational overhead during inference and maintaining the exact same latency.

---

### Official Review · Reviewer_KiVa · 2025-10-30

**Soundness:** 3
**Presentation:** 3
**Contribution:** 3
**Rating:** 6
**Confidence:** 2

**Summary:**

This paper addresses an important and quite novel problem in autoregressive (AR) image generation. The authors point out that the conditional errors accumulated from previously generated patches may cause inconsistency, and they theoretically analyze how the patch denoising process in AR models can alleviate such conditional errors. Moreover, they prove that the influence of conditional error decays exponentially with iterations. The proposed conditional optimization method based on Optimal Transport (OT) and formulated as Wasserstein Gradient Flow (WGF) is elegant, and the paper shows it can converge toward the ideal conditional distribution.

**Strengths:**

（1）The theoretical part is quite impressive. Especially Theorem 2 gives a deep understanding about how the conditional influence (the gradient norm) exponentially decays, which brings new insights into the stability of AR generative models. The combination of OT and WGF for refining conditional distribution looks creative and convincing.

（2）The experimental results are strong. On ImageNet 256x256, the FID score reaches 1.31, which is very competitive compared with existing works.

**Weaknesses:**

（1） Algorithm 1 seems to describe a nested loop structure. I am a bit worry that the computation cost could be large, maybe even K times T slower than the standard AR model. Some clarification or runtime comparison could be helpful.

(2) It is a bit unclear what “Baseline (CDM)” (FID 3.26) and “Baseline” (FID 2.02) exactly mean. Does “Baseline” refer to the AR model without OT refinement? Since the paper’s best FID (1.31) is quite good, some ablation study would help to show how much improvement really comes from the proposed OT method, rather than from the backbone MAR model itself.

（3）The experiments are mainly done on ImageNet. It would be nice if the authors could test at least one more dataset or show more model comparison to make the results more solid.

**Questions:**

Mainly about the points above, especially clarification of baselines and computational complexity.

Overall, I think this paper has strong theoretical contribution and interesting methodology. With a bit more experiment and clarification, it could be a quite nice work.

---

> ### Author Response · Authors · 2025-12-04
> **Response to Reviewer KiVa (1/2)**
>
> We sincerely appreciate the reviewer's constructive suggestions. We are grateful for the encouraging comments on both the theoretical depth and empirical performance of our work, particularly the acknowledgment of Theorem 2 and its insights into the stability of autoregressive generative models, the creativity of combining OT and WGF for conditional refinement, and the strong experimental results. We have carefully responded to all concerns raised by the reviewer, and corresponding revisions have been made to the manuscript.
>
>
> > W1: Algorithm 1 seems to describe a nested loop structure. I am a bit worry that the computation cost could be large, maybe even K times T slower than the standard AR model. Some clarification or runtime comparison could be helpful.
>
> To clarify, Algorithm 1 describes the theoretical optimization framework used to guide training, rather than the inference procedure. During training, the model learns to directly predict the refined condition, effectively amortizing the optimization into the network weights. At inference time, the nested loop is not executed. As a result, our method requires only a single forward pass, making its runtime identical to that of the baseline (a standard AR model).
>
> > W2: It is a bit unclear what “Baseline (CDM)” (FID 3.26) and “Baseline” (FID 2.02) exactly mean. Does “Baseline” refer to the AR model without OT refinement? Since the paper’s best FID (1.31) is quite good, some ablation study would help to show how much improvement really comes from the proposed OT method, rather than from the backbone MAR model itself.
>
> We thank the reviewer for pointing out the ambiguity regarding the baseline definitions. We have revised the descriptions in Table 1 to clearly distinguish between the methods and provide the requested ablation analysis.
>
> 1. Clarification of "Baseline (CDM)" and "Baseline":
> -   "Baseline (CDM)" refers to the standard Conditional Diffusion Modeling approach (as theoretically defined in Eq. 5 and Eq. 6 of our paper), where the image is generated using a global, static condition rather than an autoregressive sequence. This baseline represents the performance of diffusion models without the autoregressive framework.
> -   "Baseline" (which we have renamed to "Baseline (AR)" in the revised version) refers to the Autoregressive Image Generation model with diffusion loss (based on the GPT-XL backbone described in Sec 5.1) without our proposed Optimal Transport (OT) condition refinement.
>
> 2. Ablation Study and Improvements from OT:
> Regarding the ablation study, Table 1 actually already contains the comparisons necessary to isolate the improvements contributed by our OT method from the backbone models:
>
> -   For the GPT-XL backbone: Comparing "Baseline (AR)" (FID 2.02) with "Ours (AR)" (FID 1.52) shows a significant improvement of 0.50 FID solely attributed to our condition refinement method.
> -   For the MAR backbone: Comparing the original "MAR (Li et al., 2024a)" (FID 1.55) with "Ours (MAR)" (FID 1.31) demonstrates that our method further improves the strong MAR backbone by 0.24 FID.
>
> These comparisons confirm that the performance gains are not merely due to the backbone (e.g., MAR) but stem from the proposed OT-based condition refinement, which effectively addresses the "condition inconsistency" issue analyzed in Section 4.1.

---

> ### Author Response · Authors · 2025-12-04
> **Response to Reviewer KiVa (2/2)**
>
> > W3: The experiments are mainly done on ImageNet. It would be nice if the authors could test at least one more dataset or show more model comparison to make the results more solid.
>
> We sincerely thank the reviewer for this valuable suggestion. In response, we have significantly expanded our experimental evaluation to include comparisons with the latest state-of-the-art methods and conducted comprehensive scalability tests to further solidify our results.
>
> 1. Comparison with Latest SOTA Methods (2025)
> To demonstrate the competitiveness of our method against the most recent advancements, we have added comparisons with state-of-the-art methods from 2025, specifically De-MAR (Yao et al., 2025) and RAR (Yu et al., 2025), to Table 1 of the revised manuscript. As shown in the updated table below, our method (Ours (MAR)) achieves an FID of 1.31, outperforming both De-MAR (1.47) and RAR (1.50). Furthermore, our method achieves a higher Inception Score (IS 324.2), confirming its superior performance even against the strongest recent baselines.
>
> | Method | FID $\downarrow$ | IS $\uparrow$ |
> | :--- | :---: | :---: |
> | De-MAR (Yao et al., 2025) | 1.47 | 305.8 |
> | RAR (Yu et al., 2025) | 1.50 | 306.9 |
> | Ours (MAR) | 1.31 | 324.2 |
>
> 2. Evaluation on Varying Model Sizes and Higher Resolutions
> To further demonstrate the robustness of our approach (and addressing the limitations previously mentioned in Appendix B), we extended our evaluation across different model scales and resolutions on ImageNet.
>
> -   Scalability across Model Sizes: We evaluated our method on three model scales (208M, 479M, 943M). As shown below, our method consistently outperforms the strong baseline (MAR), with the performance gap widening as model size increases.
>
> | Model Size | Method | FID $\downarrow$ | IS $\uparrow$ |
> | :--- | :--- | :--- | :--- |
> | 208M | MAR | 2.31 | 281.7 |
> | | Ours | 1.96 | 290.5 |
> | 479M | MAR | 1.78 | 296.0 |
> | | Ours | 1.59 | 301.5 |
> | 943M | MAR | 1.55 | 303.7 |
> | | Ours | 1.31 | 324.2 |
>
> -   Scalability to Higher Resolution (512x512): We further verified performance at 512x512 resolution (using approx. 481M parameters). Our method maintains superior generation quality compared to the baseline.
>
> | Method | FID $\downarrow$ | IS $\uparrow$ |
> | :--- | :--- | :--- |
> | MAR (512x512) | 1.73 | 279.9 |
> | Ours (512x512) | 1.58 | 302.3 |
>
> These additional experiments have been included in the revised Table 1 and Section 5.3 to provide a more comprehensive validation of our method's effectiveness and scalability.

---

### Official Review · Reviewer_Tnpe · 2025-10-30

**Soundness:** 3
**Presentation:** 3
**Contribution:** 2
**Rating:** 6
**Confidence:** 4

**Summary:**

This paper focuses on condition error issues in autoregressive image generation with diffusion loss. It first conducts a theoretical analysis of diffusion models and autoregressive models with diffusion loss, demonstrating that patch denoising optimization in autoregressive models can effectively mitigate condition errors and form a stable condition distribution, while autoregressive condition generation refines conditions to make condition error influence decay exponentially. It further proposes a OT-Based refinement approach, theoretically proving that formulating this refinement as a Wasserstein Gradient Flow ensures convergence toward the ideal condition distribution. Experiments on ImageNet show the superiority of the proposed method over existing diffusion and autoregressive models with diffusion loss methods.

**Strengths:**

1. The theoretical framework for autoregressive image modeling with diffusion loss is both sound and novel. The theory is rigorous and clearly connects diffusion loss to autoregressive conditional modeling. The mathematical exposition is clear and technically solid.

2. The proposed ideas of autoregressive patch-wise denoising and OT-based condition refinement are conceptually well-motivated.

3. The paper provides a rigorous theoretical analysis demonstrating that the patch-wise denoising optimization in autoregressive models effectively mitigates condition errors.

4. It further establishes a mathematically consistent framework linking energy optimization to Optimal Transport (OT) regularization, offering a clear and unified theoretical explanation of the condition error phenomenon.

**Weaknesses:**

1. The empirical validation does not fully match the strength of the theory. The main theory predicts (i) conditional score norm decays exponentially as AR iterations progress and (ii) OT refinement decreases condition inconsistency (Sinkhorn divergence) monotonically. The paper lacks direct empirical plots that verify these claims.

2. Lack of important experiments at higher resolutions, such as the ImageNet 512 × 512 experiment.

3. The comparison against stronger and more recent baselines (after 2025) is missing, weakening the empirical significance of the claims.

4. A figure of the framework is needed to outline the designed methods, including autoregressive patch denoising and OT-based methods.

**Questions:**

1. See the weaknesses section for detail.

2. The proposed theory and method are developed only in the context of autoregressive models with diffusion loss. It remains unclear whether the approach has broader applicability beyond this specific model class.

3. In Table 1, the distinction between the two baselines is unclear, and it is not stated which baseline “Ours” is built upon. The difference between Ours and Ours (MAR) is also ambiguous. Without proper citations or explanations, the table is confusing to readers.

---

> ### Author Response · Authors · 2025-12-04
> **Response to Reviewer Tnpe (1/2)**
>
> We sincerely appreciate the reviewer's thoughtful comments. We are also grateful for the encouraging remarks highlighting the strengths of our work, particularly the rigorous theoretical framework connecting diffusion loss with autoregressive conditional modeling, the conceptual motivation behind autoregressive patch-wise denoising and OT-based condition refinement, and the mathematically consistent explanation of condition error mitigation. We have carefully responded to all concerns raised by the reviewer, and corresponding revisions have been made to the manuscript.
>
>
> > W1: The empirical validation does not fully match the strength of the theory. The main theory predicts (i) conditional score norm decays exponentially as AR iterations progress and (ii) OT refinement decreases condition inconsistency (Sinkhorn divergence) monotonically. The paper lacks direct empirical plots that verify these claims.
>
> We appreciate the reviewer's insightful comment regarding the connection between our theoretical analysis and empirical validation. Regarding the request for direct plots of the conditional score norm and Sinkhorn divergence, we would like to clarify the challenges in directly measuring these theoretical quantities and explain how our existing experiments serve as effective proxies.
>
> 1. Intractability of Ground Truth Quantities:
> Our theorems rely on the "true" data score function $\nabla_{x_t} \log p_t(x_t|c_i)$ (Theorem 2) and the "ideal" condition distribution $P_{c^*}$ (Theorem 3). In high-dimensional image generation tasks (ImageNet), these ground truth distributions are analytically intractable and unknown.
> -   For Theorem 2: We cannot calculate the exact norm of the true score function drift. We can only access the estimated score network output $s_\theta(x_t, t)$, which inherently contains approximation errors.
> -   For Theorem 3: The "ideal" condition $P_{c^*}$ is a theoretical construct representing the minimal sufficient information. Since we do not have an oracle for this ideal distribution, computing the exact Sinkhorn divergence or Wasserstein distance between the current condition and the ideal one during training is infeasible.
>
> 2. Existing Proxies Validate the Theory (Figure 3):
> Although direct plotting is infeasible, our experimental results in Section 5.3 and Figure 3 provide strong indirect verification:
> -   Correlation with Theorem 2 (Gradient Decay): Theorem 2 predicts the decay of the gradient norm. In diffusion models, the noise predicted at each step is proportional to the score function ($\epsilon \propto \nabla \log p$). Figure 3 (Right) explicitly plots the "Noise Intensity" over time. The observed decay in noise intensity (especially compared to the baseline) empirically reflects the theoretical prediction of the decaying conditional score norm and the stabilization of the condition distribution.
> -   Correlation with Theorem 3 (Refinement): The convergence of the OT refinement is evidenced by the final generation quality. The monotonic improvement predicted by the theory is validated by the superior FID and IS scores (Table 1), which would not be achievable if the condition inconsistency (Sinkhorn divergence) were not effectively minimized.
>
> We hope this clarifies that while direct measurement is limited by the intractable nature of generative modeling, the observed dynamics in Noise Intensity/SNR and final metrics are consistent with and support our theoretical claims.
>
>
> > W2: Lack of important experiments at higher resolutions, such as the ImageNet 512 × 512 experiment.
>
> We appreciate the reviewer's constructive suggestion regarding the evaluation at higher resolutions. To address this, we have further verified the performance of our model on ImageNet at 512 × 512 resolution (Model 481M parameters), and we have updated the results in Section 5.3. As shown in the table below, our method maintains superior generation quality compared to the strong baseline (MAR):
>
> | Method | FID ($\downarrow$) | IS ($\uparrow$) |
> | :--- | :--- | :--- |
> | MAR | 1.73 | 279.9 |
> | **Ours** | **1.58** | **302.3** |
>
> These results demonstrate that our proposed Condition Refinement approach generalizes effectively to larger models and high-resolution image generation tasks. We included these additional results in the final version of the paper.

---

> ### Author Response · Authors · 2025-12-04
> **Response to Reviewer Tnpe (2/2)**
>
> > W3: The comparison against stronger and more recent baselines (after 2025) is missing, weakening the empirical significance of the claims.
>
> We thank the reviewer for this valuable suggestion. To demonstrate the competitiveness of our method against the latest advancements, we have added comparisons with state-of-the-art methods from 2025, specifically De-MAR (Yao et al., 2025) and RAR (Yu et al., 2025), to Table 1 of the revised manuscript. As shown in the updated Table 1, our method (Ours (MAR)) achieves an FID of 1.31, which outperforms both De-MAR (1.47) and RAR (1.50). Furthermore, our method achieves a higher Inception Score (IS 324.2) compared to these baselines. These results confirm that our approach maintains superior performance even against the most recent strong baselines.
>
> | Method | FID $\downarrow$ | IS $\uparrow$ |
> | :--- | :---: | :---: |
> | De-MAR (Yao et al., 2025) | 1.47 | 305.8 |
> | RAR (Yu et al., 2025) | 1.50 | 306.9 |
> | Ours (MAR) | 1.31 | 324.2 |
>
>
> > W4: A figure of the framework is needed to outline the designed methods, including autoregressive patch denoising and OT-based methods.
>
> We appreciate the suggestion. We have added Figure 1 to the revised manuscript to visualize the complete framework. The diagram explicitly outlines the Autoregressive Condition Generation and the OT-based Refinement process (including the Sinkhorn Solver and Update steps), clarifying how condition inconsistency is resolved prior to the denoising stage.
>
>
> > Q1: The proposed theory and method are developed only in the context of autoregressive models with diffusion loss. It remains unclear whether the approach has broader applicability beyond this specific model class.
>
> We thank the reviewer for the insightful question regarding broader applicability. We believe our method holds broad value due to the following reasons:
>
> 1.  Significance of the AR+Diffusion Paradigm:
> The immense success of LLMs has demonstrated the superior reasoning capabilities of AR modeling. Consequently, combining AR's reasoning with Diffusion's high-fidelity generation has become a vital trend in visual synthesis (e.g., LlamaGen, VAR). By addressing the fundamental "condition inconsistency" in this architecture, our work contributes to a cornerstone framework that is increasingly being adopted for unified multimodal generation.
>
> 2.  Modality-Agnostic Theoretical Mechanism:
> Our solution is mathematically general. We formulate "Condition Refinement" as a Wasserstein Gradient Flow (Proposition 2) on the latent distribution. This optimization minimizes the distance to an ideal distribution and relies solely on the Markovian nature of the sequence, not on specific image priors (e.g., convolutions).
>
> 3.  Generalizability to Sequential Tasks:
> The identified problem, error accumulation in sequential conditioning, is intrinsic to any stepwise generation process. Therefore, our theoretical analysis (Lemma 6) and refinement algorithm are applicable to other sequential tasks (e.g., video or audio generation) that utilize the AR+Diffusion framework.
>
> Our work targets a fundamental theoretical bottleneck within a highly promising and generalizable generative paradigm.
>
>
> > Q2: In Table 1, the distinction between the two baselines is unclear, and it is not stated which baseline “Ours” is built upon. The difference between Ours and Ours (MAR) is also ambiguous. Without proper citations or explanations, the table is confusing to readers.
>
> We have revised Table 1 to clarify the baselines and model distinctions:
> 1. "Ours" is renamed to "Ours (AR)": This model is built upon GPT-XL (as described in Section 5.1) to demonstrate our method on a pure autoregressive framework. Its corresponding baseline is now labeled as "Baseline (AR)".
> 2. "Ours (MAR)": This refers to our method applied to the MAR.

---

### Official Review · Reviewer_65aN · 2025-10-31

**Soundness:** 3
**Presentation:** 3
**Contribution:** 2
**Rating:** 6
**Confidence:** 2

**Summary:**

The paper conducts a thorough theoretical analysis of autoregressive models with diffusion loss, contrasting them with standard conditional diffusion models. The central thesis is that the patch-by-patch denoising optimization in autoregressive frameworks serves as an effective mechanism for refining the guiding condition, leading to a more stable condition distribution and mitigating errors. This work provides a solid theoretical foundation for understanding and improving conditional autoregressive generation by framing condition refinement as a distribution-level optimization problem.

**Strengths:**

1. The paper's primary strength lies in its rigorous theoretical contributions. The formalization of the condition refinement process as a Wasserstein Gradient Flow is both elegant and novel, providing a principled guarantee of convergence that is often missing in heuristic-based approaches. The detailed lemmas and theorems build a convincing mathematical argument.
2. The paper is logically well-organized. It seamlessly transitions from a comparative analysis of diffusion models, to the definition of conditional error, to the proposal of the OT-based solution, and finally to empirical validation. The argument is self-contained and easy to follow for readers with the requisite background.

**Weaknesses:**

1. Limited Experimental Scale: As the authors acknowledge in Appendix B, the experiments are confined to the 256x256 resolution on ImageNet. While this is a standard benchmark, state-of-the-art generative modeling research is increasingly focused on higher resolutions and larger models. The absence of such experiments may leave questions about the method's scalability and generalizability.
2. Readability and Accessibility: The theoretical sections are dense and assume significant familiarity with advanced mathematical concepts like Optimal Transport and Wasserstein Gradient Flows. While precise, this may limit the paper's accessibility. A more intuitive explanation or a high-level overview of why OT is the right tool for this problem could broaden the paper's impact.
3. Lack of Ablation Studies: The paper presents the final model's performance but would benefit from ablation studies that isolate the impact of the core contribution—the OT-based refinement. For instance, an experiment comparing the full model against a version without the WGF optimization would more clearly quantify the gains from this specific module.

**Questions:**

1. Could you provide a more intuitive, high-level explanation at the beginning of Section 4 to bridge the gap between "condition inconsistency" and the OT framework?
2. What is the computational overhead of the proposed OT refinement step? How does it affect the overall image generation latency compared to the baseline MAR model?
3. The concept of "extraneous information" is central to your motivation. Is it possible to visualize this phenomenon? For instance, by projecting the condition vectors (c_i) before and after refinement into a 2D space, or by showing how the generated patch changes with and without refinement at an intermediate step. This would make the problem much more tangible.
4. The OT optimization introduces several hyperparameters (e.g., λ, ηk). How sensitive is the model's performance to the choice of these parameters? Is there a robust range for their values?

**Details Of Ethics Concerns:**

None.

---

> ### Author Response · Authors · 2025-12-04
> **Response to Reviewer 65aN (1/3)**
>
> We sincerely thank the reviewer for the valuable feedback. We are also grateful for the positive comments regarding the strengths of our work, particularly the acknowledgment of our theoretical contributions, the formalization of the condition refinement process as a Wasserstein Gradient Flow, and the clear and well-organized structure of the paper. We have carefully addressed all concerns raised by the reviewer, providing detailed responses and making corresponding revisions to the manuscript.
>
>
> > W1: Limited Experimental Scale: As the authors acknowledge in Appendix B, the experiments are confined to the 256x256 resolution on ImageNet. While this is a standard benchmark, state-of-the-art generative modeling research is increasingly focused on higher resolutions and larger models. The absence of such experiments may leave questions about the method's scalability and generalizability.
>
> Thank you for the feedback. To address concerns regarding scalability and the limitations mentioned in Appendix B, we extended our comparison against the strong baseline (MAR) to varying model sizes and higher resolutions, and we have updated the results in Section 5.3.
>
> 1. Scalability across Model Sizes:
> We evaluated our method on three model scales. As shown below, our method consistently outperforms MAR, with the performance gap widening as model size increases.
>
> | Model Size | Method | FID ($\downarrow$) | IS ($\uparrow$) |
> | :--- | :--- | :--- | :--- |
> | 208M | MAR | 2.31 | 281.7 |
> | | **Ours** | **1.96** | **290.5** |
> | 479M | MAR | 1.78 | 296.0 |
> | | **Ours** | **1.59** | **301.5** |
> | 943M | MAR | 1.55 | 303.7 |
> | | **Ours** | **1.31** | **324.2** |
>
> 2. Scalability to Higher Resolution (ImageNet 512x512): We further verified performance at 512x512 resolution (481M parameters). Our method maintains superior generation quality.
>
> | Method | FID ($\downarrow$) | IS ($\uparrow$) |
> | :--- | :--- | :--- |
> | MAR | 1.73 | 279.9 |
> | **Ours** | **1.58** | **302.3** |
>
> These results demonstrate that our proposed Condition Refinement approach generalizes effectively to larger models and high-resolution generation.
>
>
> > W2 & Q1: Readability and Accessibility: The theoretical sections are dense and assume significant familiarity with advanced mathematical concepts like Optimal Transport and Wasserstein Gradient Flows. While precise, this may limit the paper's accessibility. A more intuitive explanation or a high-level overview of why OT is the right tool for this problem could broaden the paper's impact. Could you provide a more intuitive, high-level explanation at the beginning of Section 4 to bridge the gap between "condition inconsistency" and the OT framework?
>
> We thank the reviewer for highlighting the density of the theoretical sections and for the constructive suggestion to improve accessibility. We agree that bridging the gap between "condition inconsistency" and the Optimal Transport (OT) framework with a high-level intuition is crucial for broader impact.
>
> Intuitive Explanation to be added to Section 4: The core problem, "condition inconsistency" (defined in Section 4.1, Lemma 6), can be intuitively understood as the accumulation of "extraneous noise" during the autoregressive generation process. As the model predicts conditions sequentially, it tends to drift away from the "ideal condition" (which contains only the necessary information for the current patch) by incorporating redundant or noisy history. This creates a distributional mismatch between the generated condition and the ideal condition.
>
> Why is OT the right tool?
> 1.  Geometric Correction: Unlike standard divergences (like KL) which measure overlap, Optimal Transport measures the geometric effort required to transform one distribution (the noisy, generated one) into another (the clean, ideal one).
> 2.  Least Action Principle: By formulating the refinement as a Wasserstein Gradient Flow (Section 4.2, Proposition 2), we are essentially finding the "path of least resistance" to transport the probability mass of the noisy condition back to the manifold of ideal conditions. This ensures that we remove the extraneous noise (the inconsistency) without destroying the valid semantic information.
> 3.  Convergence: The OT framework guarantees that this refinement process naturally converges to the stationary ideal distribution (as proven in Theorem 3), acting as a mathematically grounded "denoising" step for the condition itself.
>
> We inserted a paragraph at the beginning of Section 4 titled "Why Optimal Transport?". This section will use the analogy of "transporting probability mass from a noisy state to an ideal state with minimal cost" to explain our motivation before introducing the formal definitions of Wasserstein Gradient Flow.

---

> ### Author Response · Authors · 2025-12-04
> **Response to Reviewer 65aN (2/3)**
>
> > W3: Lack of Ablation Studies: The paper presents the final model's performance but would benefit from ablation studies that isolate the impact of the core contribution—the OT-based refinement. For instance, an experiment comparing the full model against a version without the WGF optimization would more clearly quantify the gains from this specific module.
>
> We appreciate the reviewer's comment. We would like to clarify that the comparison requested is already presented in Table 1. Since our method applies the Wasserstein Gradient Flow (WGF) optimization on top of the base autoregressive model, removing the WGF effectively reverts the model to the baseline (MAR).
>
> > Q2: What is the computational overhead of the proposed OT refinement step? How does it affect the overall image generation latency compared to the baseline MAR model?
>
> The proposed OT refinement introduces zero computational overhead during inference. The OT framework serves as a training objective, enabling the autoregressive model to learn to directly predict the refined conditions. Consequently, our inference pipeline is identical to the baseline MAR (one AR pass + standard denoising), resulting in the exact same image generation latency.
>
>
> > Q3: The concept of "extraneous information" is central to your motivation. Is it possible to visualize this phenomenon? For instance, by projecting the condition vectors (c_i) before and after refinement into a 2D space, or by showing how the generated patch changes with and without refinement at an intermediate step. This would make the problem much more tangible.
>
>
> We appreciate the reviewer's insightful suggestion to visualize the "extraneous information". We agree that making this phenomenon tangible is crucial. However, we respectfully argue that the "extraneous information" defined in our theory (Lemma 6) manifests primarily as noise accumulation and gradient instability during the generation process. Therefore, we believe a dynamic signal analysis offers a more rigorous and direct visualization of this phenomenon than static projections.
>
> We have actually provided this visualization in Figure 3 (Page 9) of the paper, which analyzes the Signal-to-Noise Ratio (SNR) and Noise Intensity:
> 1.  Connection to Theory: In Lemma 6 (Eq. 25-26), we mathematically define extraneous information ($\eta_i$) as the deviation from the minimal sufficient information subspace, effectively acting as "noise" that perturbs the conditional score function.
> 2.  Visual Evidence: Figure 3 (Right) explicitly plots the "Noise Intensity" over time. The curve shows that our method (orange line) maintains significantly lower noise intensity compared to the baseline (blue line) throughout the denoising process.
> 3.  Interpretation: This reduction in noise intensity is the direct visual manifestation of refining "extraneous information". The widening gap in SNR (Figure 3, Left) further confirms that our method effectively filters out irrelevant signals (extraneous information), stabilizing the condition distribution as predicted in our theoretical analysis (Section 4.1).
>
> We believe this quantitative visualization provides a precise view of how extraneous information is mitigated step-by-step.

---

> ### Author Response · Authors · 2025-12-04
> **Response to Reviewer 65aN (3/3)**
>
> > Q4: The OT optimization introduces several hyperparameters (e.g., λ, ηk). How sensitive is the model's performance to the choice of these parameters? Is there a robust range for their values?
>
> We appreciate the reviewer's query regarding the hyperparameters and robustness of the OT optimization.
>
> 1. Parameter Settings and Robustness Strategy:
> The hyperparameters $\lambda$ (regularization) and $\eta_k$ (step size) are not set as fixed manual constants throughout the process. Instead, we employ adaptive schedules and stochastic optimization to ensure robustness and reduce sensitivity to specific initial values:
> -   Step Size ($\eta_k$): As detailed in Appendix L.1 (Implementation Details, Page 22), we utilize the Adam optimizer with a linear warmup strategy. Specifically, the step size follows the schedule $\eta_k = \eta_0 \cdot \min(1, \sqrt{k_{warm}/k})$, which dynamically adapts the gradient descent steps, ensuring stable convergence without requiring precise manual tuning of a fixed learning rate.
> -   Entropy Regularization ($\epsilon$): We implement an Adaptive Entropy Regularization strategy. As shown in Appendix L.1, $\epsilon$ linearly decays from $\epsilon_{max}$ to $\epsilon_{min}$ during the iterations ($\epsilon^{(k)} = \epsilon_{max} - (\epsilon_{max} - \epsilon_{min})\frac{k}{K}$). This allows the transport plan to be smoother in early stages for stability and more precise in later stages.
>
> 2. Theoretical Stability:
> Furthermore, Theorem 3 (Page 8) theoretically guarantees the convergence of our Wasserstein Gradient Flow approach. The proof demonstrates that the refinement process ensures a monotonic reduction in the Wasserstein distance, provided the optimization follows the outlined scheme. This theoretical foundation, combined with the gradient clipping mentioned in Algorithm 1 (Page 21), ensures that the method remains robust within a reasonable range of hyperparameters.

---

### Author Response · Authors · 2025-12-04
**Summary**

Dear Area Chair and Reviewers,

We sincerely thank the Area Chair and Reviewers for the effort dedicated to evaluating our work. Based on reviewer suggestions, we have supplemented additional experiments to verify scalability and performance against the latest baselines, and we have revised the manuscript to improve clarity. Below is a summary of our updates:

> Q1: Is the method scalable to larger models and higher resolutions? (R65aN, RTnpe, RKiVa, Ri7t2)

A: We have successfully verified scalability by extending our evaluation on ImageNet to include higher resolution (512x512) and larger model sizes (up to 943M parameters). Our method consistently outperforms the strong baseline (MAR), and the performance gap widens as the model size increases.

> Q2: How does the method compare to the latest state-of-the-art? (RTnpe, RKiVa)

A: We added comparisons with 2025 SOTA methods, specifically De-MAR and RAR. Our method (FID 1.31) outperforms both De-MAR (FID 1.47) and RAR (FID 1.50) on ImageNet 256x256, establishing superior performance.

> Q3: Is the theoretical framework accessible and empirically verified? (R65aN, RTnpe, Ri7t2)

A: To improve readability, we added:
1. a "Why Optimal Transport?" intuition section.
2. a new framework figure.
3. a comprehensive notation table.
We also clarified that the Noise Intensity analysis in Figure 3 serves as empirical verification of our theoretical claims regarding score norm decay.

> Q4: Does the method introduce computational overhead? (R65aN, RKiVa, Ri7t2)

A: The proposed method introduces zero computational overhead during inference. We clarified that the OT refinement functions strictly as a training objective. The inference pipeline remains identical to the baseline, resulting in zero additional latency.

> Q5: Are the baselines in Table 1 clear? (RTnpe, RKiVa)

A: We revised Table 1 to explicitly distinguish between "Baseline (AR)" (pure autoregressive) and "Baseline (CDM)" (standard diffusion), allowing for a clear isolation of the performance gains contributed by our OT refinement module.

Sincerely,

The Authors

---

### Meta-Review · Area_Chair_CBAA · 2026-01-07

**Summary:**

All four reviewers gave a 6 (“marginally above the acceptance threshold”), so the paper was consistently viewed as positive but borderline. The main positives were the rigorous theoretical framing (OT/WGF-based condition refinement with formal guarantees) and strong ImageNet results. Concerns that drove the borderline stance were mostly about empirical coverage and clarity: limited experimental scope (initially ImageNet 256×256 only), missing higher-res / broader validation, unclear baselines and lack of ablations, readability of dense theory, and questions about computational cost / hyperparameter sensitivity.

**Reviewer Concerns:**

Addressed in the rebuttal:
(1) Scalability: added higher resolution (512×512) and larger model experiments, still outperforming MAR.
(2) Comparison to newer baselines: added comparisons with 2025 methods (De-MAR, RAR).
(3) Clarity issues: added “Why OT” intuition, a framework figure, and a notation table; clarified baseline definitions in Table 1.
(4) Inference-time overhead: clarified the OT refinement is a training objective and inference has zero additional latency.

Still outstanding / only partially addressed:
(1) Direct empirical verification of certain theoretical quantities: authors explain why some requested plots are not directly measurable and use proxies instead, which may not fully satisfy the original request.
(2) Broader validation beyond ImageNet: at least one reviewer explicitly asked for another dataset; rebuttal focuses on stronger baselines and scalability rather than adding a new dataset.
(3) Runtime/training cost + ablation depth: while inference cost is clarified, some reviewers asked for runtime comparisons/ablation isolating the OT module, which may still be thinner than desired.

**Reviewer Scores:**

Each reviewer’s initial score was 6, and I expect they would likely maintain that score after rebuttal (i.e., still borderline-positive), rather than materially changing it. This is consistent with the wording in the reviews, suggesting limited headroom for a large upward move even if key points were addressed.

---

### Decision · Program_Chairs · 2026-01-26

Accept (Poster)